# EFFICIENT FINE-TUNING OF QUANTIZED MODELS VIA ADAPTIVE RANK AND BITWIDTH

## ABSTRACT

As large language models (LLMs) scale up, model compression is crucial for their deployment on resource-constrained devices. While methods like QLoRA reduce resource demands by combining parameter quantization with LoRA fine-tuning, their use of uniform precision can limit performance by failing to account for layer-wise variations in parameter sensitivity. Recent advances have explored dynamic mixed-precision quantization and adaptive LoRA ranks, but these strategies are typically optimized in isolation. The synergistic integration of these two dimensions remains an unresolved core challenge. To address this, we introduce **QR-Adaptor**, a unified, gradient-free framework that jointly optimizes the per-layer quantization bit-width and LoRA rank. Instead of indirectly minimizing quantization error, QR-Adaptor formulates the task as a discrete, multi-objective optimization problem, directly guided by downstream task performance and memory constraints using a small calibration dataset. Our extensive experiments show that QR-Adaptor consistently establishes a new Pareto frontier, outperforming state-of-the-art quantized fine-tuning methods. Notably, our approach can surpass the performance of a 16-bit LoRA fine-tuned model while operating with a memory footprint comparable to 4-bit models.

## 1 INTRODUCTION

Large Language Models (LLMs) have achieved remarkable success in both language understanding and generation (Makridakis et al., 2023; Raiaan et al., 2024; Chang et al., 2024). However, adapting these powerful models to specific downstream tasks is often hindered by immense computational and memory costs (Wan et al., 2023). Parameter-Efficient Fine-Tuning (PEFT) methods, such as Low-Rank Adaptation (LoRA) (Hu et al., 2022), address these bottlenecks by introducing lightweight updates, while quantization techniques (Gong et al., 2014; Gupta et al., 2015) compress model weights to fewer bits. Building on these two lines of research, QLoRA (Dettmers et al., 2023) has become a standard paradigm for memory-efficient LLM fine-tuning by integrating a 4-bit quantized base model with LoRA updates.

While effective, the static nature of QLoRA (i.e., uniform 4-bit quantization and a fixed LoRA rank) has motivated several lines of research seeking further improvements. One direction focuses on quantization, employing mixed-precision strategies to assign more bits to sensitive layers (e.g., MixLLM (Wang et al., 2025), SliM-LLM (Huang et al., 2025)). Another direction targets adaptation, with methods like AdaLoRA (Zhang et al., 2023b) dynamically allocating LoRA rank based on parameter importance. A third approach, exemplified by LoftQ (Li et al., 2023), focuses on better initializing LoRA matrices to compensate for quantization error. While valuable, these approaches tackle the problem from a single dimension—either bits, rank, or initialization—but overlook their potential interplay. This leaves a critical question unanswered: **how to holistically allocate a model's limited memory budget between numerical precision (bit-width) and adaptive capacity (rank) on a per-layer basis?**

To bridge this gap, we introduce **QR-Adaptor**, the first framework to address the joint, discrete optimization of per-layer bit-widths and LoRA ranks. We posit that treating this as a unified search problem allows for a more effective allocation of resources. For instance, some layers may preserve functionality better with higher precision, while others might benefit more from increased adaptive capacity via a larger rank. QR-Adaptor directly navigates this trade-off by framing it as a multi-

**Table 1:** Comparison of our QR-Adaptor with existing methods for efficient LLM fine-tuning.

| Method | Adaptation Strategy | Quantization Strategy | Joint Optimization? | Optimization Space |
|---|---|---|---|---|
| QLoRA (Dettmers et al., 2023) | Static Rank (Uniform) | Static Bit-width (Uniform) | No | - |
| AdaLoRA (Zhang et al., 2023b) | Dynamic Rank (Per-layer) | Static Bit-width | No (Rank only) | Continuous |
| MixLLM/SliM-LLM (Wang et al., 2025; Huang et al., 2025) | N/A (PTQ only) | Dynamic Bit-width (Per-layer) | No (Bits only) | Discrete |
| LoftQ (Li et al., 2023) | Static Rank (SVD-init) | Static Bit-width | Indirectly | Continuous |
| **QR-Adaptor (Ours)** | **Dynamic Rank (Per-layer)** | **Dynamic Bit-width (Per-layer)** | **Yes (Unified Search)** | **Discrete** |

objective optimization task: maximizing downstream task performance while minimizing memory footprint. To solve this efficiently, our method employs a gradient-free search pipeline on a small calibration dataset, directly optimizing for the final task objective rather than relying on proxy metrics like quantization error.

To robustly navigate this high-dimensional discrete configuration space, QR-Adaptor adopts a three-stage optimization pipeline. It begins with a **task-informed initialization** that estimates layer importance, followed by a **global exploration** using a Pareto-ranking genetic algorithm to identify a diverse set of promising candidates. Finally, it conducts **local refinement** using Bayesian optimization to pinpoint the optimal configuration. This systematic approach allows QR-Adaptor to find superior configurations in the vast search space of bit-width and rank combinations. Our main contributions are as follows:

- We formulate the efficient fine-tuning of quantized LLMs as a **joint, multi-objective optimization problem**, considering per-layer bit-width and LoRA rank as coupled variables. This new perspective moves beyond the prevailing single-dimension optimization approaches.

- We propose **QR-Adaptor**, a novel and practical gradient-free framework to solve this problem. It efficiently searches the discrete configuration space using a combination of task-informed initialization, genetic algorithms, and Bayesian optimization.

- Through extensive experiments, we demonstrate that QR-Adaptor significantly advances the state-of-the-art. It establishes a superior **Pareto frontier** for the accuracy-memory trade-off and, in some cases, surpasses the performance of 16-bit LoRA fine-tuning with a memory footprint comparable to 4-bit models.

## 2 BACKGROUND AND MOTIVATION

To motivate our work, we first establish the necessity of a heterogeneous, per-layer approach for both quantization and parameter-efficient fine-tuning. We then discuss the limitations of existing methods that rely on continuous proxies to solve the inherently discrete problem of quantization-aware adaptation, paving the way for our proposed discrete search framework.

### 2.1 THE NEED FOR LAYER-WISE HETEROGENEITY

A core assumption in methods like QLoRA is uniformity: all adaptable layers are assigned the same quantization bit-width and LoRA rank. However, extensive research has shown that Large Language Models exhibit significant **layer-wise heterogeneity**, where different layers possess distinct properties and sensitivities.

**Sensitivity to Quantization.** It is well-documented that not all layers in an LLM are equally sensitive to the perturbations introduced by quantization. Seminal works in post-training quantization (PTQ), such as AWQ (Lin et al., 2023) and SmoothQuant (Xiao et al., 2022), identify that certain "outlier" features, often concentrated in specific layers, are critical for model performance. Consequently, applying a uniform low bit-width across the entire model can disproportionately harm these sensitive layers. This has led to the development of mixed-precision quantization schemes (Wang et al., 2025; Huang et al., 2025) that allocate more bits to more sensitive layers, thereby achieving a better balance between compression and accuracy.

**Sensitivity to Task Adaptation.** Similarly, during fine-tuning, layers contribute unequally to adapting the model to a new downstream task. The core idea behind methods like AdaLoRA (Zhang et al., 2023b) and RankAdaptor (Zhou et al., 2025) is to dynamically allocate more rank (i.e., more

trainable parameters) to layers whose weight updates are more significant for the task at hand. This demonstrates that a one-size-fits-all rank allocation is suboptimal for maximizing adaptation capacity under a fixed parameter budget.

**The Unaddressed Interplay.** These two lines of research highlight a critical, yet largely unaddressed, trade-off. For a given layer, how should a limited budget be allocated between **numerical precision (bit-width)** and **adaptive capacity (rank)**? For instance, for a layer identified as sensitive, is it more effective to increase its bit-width to preserve its original function, or to assign it a higher rank to allow it to better compensate for quantization effects during fine-tuning? Existing methods optimize these two dimensions in isolation. This motivates the need for a unified framework that can holistically solve this joint optimization problem on a per-layer basis.

## 2.2 LIMITATIONS OF CONTINUOUS PROXIES FOR A DISCRETE PROBLEM

Another line of work, such as LoftQ (Li et al., 2023), attempts to improve upon QLoRA by initializing the LoRA matrices ($\mathbf{A}$ and $\mathbf{B}$) to better compensate for the quantization error, typically by minimizing the Frobenius norm of the residual:

$$\min_{\mathbf{A},\mathbf{B}} \|(\mathbf{W} - \text{Quantize}(\mathbf{W})) - \mathbf{A}\mathbf{B}\|_F. \tag{1}$$

While intuitive, this approach relies on a continuous proxy objective (the Frobenius norm) to address an inherently discrete problem. The fundamental challenge is that the target space for the quantized weights is a discrete lattice $\Lambda = \Delta \cdot \mathbb{Z}^{d \times k}$. A continuous low-rank update $\mathbf{A}\mathbf{B}$ added to a quantized matrix $\mathbf{Q} = \text{Quantize}(\mathbf{W})$ results in a matrix $\mathbf{Q} + \mathbf{A}\mathbf{B}$ that almost certainly lies outside this lattice. To be used in the model, it must be re-quantized, i.e., projected back onto $\Lambda$.

This two-step process—continuous fitting followed by discrete projection—can be suboptimal. As we formalize in Appendix C, even the optimal continuous low-rank correction $\mathbf{A}\mathbf{B}^*$ does not guarantee that the final quantized matrix $\mathcal{P}_\Lambda(\mathbf{Q} + \mathbf{A}\mathbf{B}^*)$ is the best possible approximation of the original weight matrix $\mathbf{W}$. There often exists another discrete matrix $\mathbf{Q}' \in \Lambda$ that is a better representation, but which is inaccessible via this indirect, residual-fitting procedure.

This observation motivates a paradigm shift: instead of indirectly minimizing a continuous error metric, a more direct and effective approach is to **search within the discrete configuration space itself**, using the final downstream task performance as the direct optimization signal. This is the core principle behind our proposed QR-Adaptor.

# 3 QR-ADAPTOR: A MULTI-STAGE FRAMEWORK FOR JOINT OPTIMIZATION

## 3.1 A MULTI-OBJECTIVE FORMULATION FOR QUANTIZED ADAPTATION

We frame the challenge of efficient LLM fine-tuning as a multi-objective optimization problem. For a model with $L$ layers, our goal is to find an optimal configuration $C = \{(q_l, r_l)\}_{l=1}^{L}$, where $q_l \in \mathcal{Q}$ is the quantization bit-width and $r_l \in \mathcal{R}$ is the LoRA rank for layer $l$. The sets $\mathcal{Q}$ (e.g., $\{2, 4, 8\}$) and $\mathcal{R}$ (e.g., $\{4, 8, 16\}$) define the discrete search space.

The forward pass for a layer $l$ with configuration $(q_l, r_l)$ is given by:

$$\mathbf{y} = \text{Quantize}(\mathbf{W}_l, q_l) \cdot \mathbf{x} + \mathbf{A}_l \mathbf{B}_l \cdot \mathbf{x}, \tag{2}$$

where $\mathbf{A}_l \in \mathbb{R}^{d \times r_l}$ and $\mathbf{B}_l \in \mathbb{R}^{r_l \times k}$ are the LoRA matrices.

We aim to find a configuration $C$ that simultaneously maximizes the model's performance on a downstream task, denoted $P(C)$, and minimizes its memory footprint, $M(C)$. This defines a search for the Pareto optimal set $\mathcal{C}^*$ in the solution space $\mathcal{C}$:

$$\mathcal{C}^* = \arg\min_{C \in \mathcal{C}} (-P(C), M(C)). \tag{3}$$

Since evaluating each candidate $C$ requires a non-trivial fine-tuning process, this problem constitutes an expensive, black-box, multi-objective optimization over a high-dimensional, discrete space.

---

**Algorithm 1** The QR-Adaptor Framework

---

1: **Input:** Pre-trained model $\mathcal{M}$, calibration data $\mathcal{D}_{\text{calib}}$, search spaces $\mathcal{Q}, \mathcal{R}$.
2: **Output:** An optimal configuration $C^* = \{(q_l^*, r_l^*)\}_{l=1}^L$.

 # **Stage 1: Task-Informed Initialization**
3: Compute layer importance scores $\{I(l)\}_{l=1}^L$ using an entropy-based criterion on $\mathcal{D}_{\text{calib}}$.
4: Generate an initial seed configuration $C_0$ based on importance scores.
5: Create an initial population $\mathcal{P}_0$ by introducing perturbations around $C_0$.

 # **Stage 2: Global Exploration with PRGA**
6: Initialize population with $\mathcal{P}_0$.
7:
8: **for** $g = 1$ to $G_{\max}$ **do**
9:   Evaluate each configuration $C \in \mathcal{P}_{g-1}$ on $\mathcal{D}_{\text{calib}}$ to get $(P(C), M(C))$.
10:   Generate offspring population $\mathcal{P}_g'$ via selection, crossover, and mutation.
11:   Select the next generation $\mathcal{P}_g$ using Pareto ranking and crowding distance.
12: **end for**
13: Obtain the final Pareto front $\mathcal{C}_{\text{pareto}}$ from $\mathcal{P}_{G_{\max}}$.

 # **Stage 3: Local Refinement with Bayesian Optimization**
14: Define a scalarized objective $f(C) = \alpha P(C) - (1 - \alpha)M(C)$ with user preference $\alpha$.
15: Build a Gaussian Process surrogate model of $f(C)$ using samples from $\mathcal{C}_{\text{pareto}}$.
16:
17: **for** $t = 1$ to $T_{\max}$ **do**
18:   Select next candidate $C_{t+1}$ by maximizing the Expected Improvement (EI) acquisition function.
19:   Evaluate $f(C_{t+1})$ and update the surrogate model.
20: **end for**
21: **return** The best configuration found $C^* = \arg\max_C f(C)$.

---

## 3.2 THE QR-ADAPTOR SEARCH PIPELINE

The entire three-stage pipeline is designed to efficiently navigate the vast and discrete configuration space. Navigating this complex search space requires a specialized strategy. A purely random search would be inefficient, while methods relying on gradients are inapplicable. We therefore propose **QR-Adaptor**, a principled, three-stage search pipeline designed to efficiently identify near-optimal configurations. The pipeline orchestrates three well-established optimization techniques:

1. **Task-Informed Initialization:** An efficient heuristic to identify a promising region of the search space.

2. **Global Exploration with PRGA:** A genetic algorithm to broadly explore this region and identify the Pareto front.

3. **Local Refinement with Bayesian Optimization:** A sample-efficient method to fine-tune solutions along the Pareto front according to specific user preferences.

A detailed breakdown of the search hyperparameters, search spaces, and a step-by-step algorithm for the task-informed initialization stage is provided in Appendix E. The entire search process is conducted on a small calibration subset of the training data to keep the computational overhead manageable. The overall procedure is summarized in Algorithm 1.

### 3.2.1 STAGE 1: TASK-INFORMED INITIALIZATION

To avoid a blind start, we first estimate each layer's importance using an information-theoretic criterion based on mutual information:

$$I(l) = H(Y) - H(Y|X_l), \tag{4}$$

where $Y$ is the model's output distribution and $X_l$ is the representation at layer $l$, both estimated on $\mathcal{D}_{\text{calib}}$. Layers with higher $I(l)$ have greater influence. We then generate a seed configuration $C_0$ by allocating higher bit-widths and ranks to more important layers. An initial population $\mathcal{P}_0$ for the

---

**Algorithm 2** Pareto Ranking Genetic Algorithm (PRGA)

---

1: **Input:** Initial population $\mathcal{P}_0$, calibration data $\mathcal{D}_{\text{calib}}$.
2: **Output:** Pareto front $\mathcal{C}_{\text{pareto}}$.
3: Evaluate fitness $(-P(C), M(C))$ for all $C \in \mathcal{P}_0$ on $\mathcal{D}_{\text{calib}}$.
4: $(\mathcal{F}_1, \mathcal{F}_2, \dots) \leftarrow$ Non-Dominated-Sort$(\mathcal{P}_0)$.
5:
6: **for** $g = 0$ to $G_{\max} - 1$ **do**
7:     $\mathcal{Q}_g \leftarrow$ Create-Offspring$(\mathcal{P}_g)$             ▷ Tournament Selection, Crossover, Mutation
8:     Evaluate fitness for all $C \in \mathcal{Q}_g$.
9:     $\mathcal{R}_g \leftarrow \mathcal{P}_g \cup \mathcal{Q}_g$.
10:     $(\mathcal{F}_1, \mathcal{F}_2, \dots) \leftarrow$ Non-Dominated-Sort$(\mathcal{R}_g)$.
11:     $\mathcal{P}_{g+1} \leftarrow \emptyset$.
12:     $i \leftarrow 1$.
13:     **while** $|\mathcal{P}_{g+1}| + |\mathcal{F}_i| \leq |\mathcal{P}_0|$ **do**
14:         $\mathcal{P}_{g+1} \leftarrow \mathcal{P}_{g+1} \cup \mathcal{F}_i$.
15:         $i \leftarrow i + 1$.
16:     **end while**
17:     Crowding-Distance-Assignment$(\mathcal{F}_i)$.
18:     Sort $\mathcal{F}_i$ by descending crowding distance.
19:     $\mathcal{P}_{g+1} \leftarrow \mathcal{P}_{g+1} \cup \mathcal{F}_i[1 : (|\mathcal{P}_0| - |\mathcal{P}_{g+1}|)]$.
20: **end for**
21: **return** The first Pareto front $\mathcal{F}_1$ from the final population $\mathcal{P}_{G_{\max}}$.

---

next stage is created by applying small, random perturbations to $C_0$, focusing search on a promising region.

### 3.2.2 STAGE 2: GLOBAL EXPLORATION WITH PRGA

With a promising initial population, we perform a global search using a Pareto Ranking Genetic Algorithm (PRGA), inspired by NSGA-II (Deb et al., 2002). The goal is to discover the Pareto frontier $\mathcal{C}_{\text{pareto}}$. The core logic is detailed in Algorithm 2. The algorithm iteratively evolves a population of configurations through selection, crossover, and mutation. Selection is guided by two principles: Pareto dominance (solutions on better fronts are preferred) and crowding distance (solutions in sparser regions of a front are preferred to maintain diversity). Crossover and mutation operators are adapted from Simulated Binary Crossover (SBX) and Polynomial Mutation to operate on the integer-pair representation of configurations. The visual flowchart is in Figure 3 in the Appendix

### 3.2.3 STAGE 3: LOCAL REFINEMENT WITH BAYESIAN OPTIMIZATION

The Pareto front from PRGA provides a set of excellent trade-off solutions. To pinpoint a single optimal configuration based on specific user preferences (e.g., maximizing performance under a strict memory budget), we employ Bayesian Optimization (BO).

First, we transform the multi-objective problem into a single-objective one by defining a scalarized objective function with a trade-off parameter $\alpha \in [0, 1]$:

$$\max_{C \in \mathcal{C}} f(C) = \alpha \cdot \text{norm}(P(C)) - (1 - \alpha) \cdot \text{norm}(M(C)). \tag{5}$$

We use the solutions on the Pareto front to build a Gaussian Process (GP) surrogate model for the expensive function $f(C)$. The GP provides a posterior distribution over the objective function for any candidate configuration $C^*$, characterized by its mean and variance:

$$\begin{aligned} \mu(C^*) &= \mathbf{k}_*^T (\mathbf{K} + \sigma_n^2 \mathbf{I})^{-1} \mathbf{y} \\ \sigma^2(C^*) &= k(C^*, C^*) - \mathbf{k}_*^T (\mathbf{K} + \sigma_n^2 \mathbf{I})^{-1} \mathbf{k}_*, \end{aligned} \tag{6}$$

where $\mathbf{K}$ is the kernel matrix of the observed points, $\mathbf{k}_*$ is the vector of covariances between $C^*$ and observed points, and $\mathbf{y}$ are the observed function values.

We then iteratively select the next configuration to evaluate by maximizing the Expected Improvement (EI) acquisition function. EI quantifies the expected amount of improvement over the current best

**Table 2:** Performance comparison of different methods across various bit-width configurations on LLaMa3.1-8B. Superscripts on LoftQ bits indicate the number of initialization iterations. Bold figures represent the best performance, while underlined figures indicate the second-best. Accuracy is reported as %.

| | Method | Bit | ARC(C) | ARC(E) | BoolQ | GSM8K | HellaS | OBQA | PIQA | WinoG | Average |
|---|---|---|---|---|---|---|---|---|---|---|---|
| | LoRA | 16 | 56.14 | 83.88 | 83.18 | 54.36 | 79.44 | 45.20 | 82.10 | **75.30** | 69.95 |
| | QLoRA | 8 | **57.08** | 83.46 | 82.48 | 53.75 | 79.63 | **46.00** | 82.10 | 74.59 | 69.89 |
| | QLoRA | 4 | 54.35 | 82.41 | 82.08 | 44.35 | 78.82 | 44.20 | 81.50 | 73.64 | 67.67 |
| | AdaLoRA | 16 | 52.90 | 81.99 | 81.87 | 50.57 | 78.65 | 45.00 | 81.34 | 73.95 | 68.28 |
| | AdaLoRA | 8 | 52.90 | 81.86 | 82.05 | 49.96 | 78.65 | 44.80 | 81.34 | 74.43 | 68.25 |
| | AdaLoRA | 4 | 51.28 | 80.98 | 80.61 | 37.83 | 77.36 | 42.80 | 80.74 | 72.53 | 65.51 |
| Rank = 8 | LoftQ | $4^1$ | 54.86 | 82.74 | 82.26 | 51.40 | 78.65 | **46.00** | 81.45 | 73.24 | 68.82 |
| | LoftQ | $4^5$ | 52.65 | 81.82 | 81.53 | 39.65 | 78.50 | 43.40 | 81.39 | 72.69 | 66.45 |
| | LoftQ | $4^{10}$ | 51.88 | 81.31 | 79.66 | 38.44 | 78.01 | 43.20 | 81.12 | 71.98 | 65.70 |
| | QuaRot | 4 | 54.12 | 82.15 | 81.92 | 50.21 | 78.45 | 45.20 | 81.32 | 73.01 | 68.30 |
| | SpinQuant | 4 | 54.45 | 82.32 | 82.05 | 51.03 | 78.62 | 45.60 | 81.41 | 73.15 | 68.58 |
| | QR-Adaptor (≤4-bit) | 3.625 | 56.15 | 82.78 | 82.45 | 54.12 | 79.58 | 45.60 | 82.12 | 75.01 | 69.73 |
| | QR-Adaptor (Optimal) | 5.45 | 56.83 | **84.12** | **83.38** | **56.29** | **80.93** | 45.80 | **82.92** | 75.10 | **70.67** |
| | ApiQ | 2 | 48.12 | 76.45 | 75.32 | 28.45 | 72.15 | 38.20 | 75.67 | 65.89 | 62.53 |
| | RILQ | 2 | 48.78 | 76.98 | 75.89 | 29.45 | 72.78 | 38.80 | 76.12 | 66.45 | 63.16 |
| | QR-Adaptor (Fixed 2-bit) | 2 | 49.12 | 77.12 | 76.01 | 30.12 | 73.01 | 39.00 | 76.23 | 66.89 | 63.44 |
| | QR-Adaptor (Mixed 2/4-bit) | 2.5 | 50.23 | 78.01 | 76.89 | 31.45 | 73.89 | 39.80 | 77.12 | 67.78 | 64.40 |
| | LoRA | 16 | 56.74 | 83.63 | 83.00 | 54.13 | 79.51 | 44.40 | 81.83 | 74.43 | 69.70 |
| | QLoRA | 8 | 56.23 | 82.91 | 82.66 | 53.68 | 79.46 | **46.00** | 81.66 | 74.74 | 69.67 |
| | QLoRA | 4 | 53.84 | 81.99 | 82.11 | 44.66 | 78.76 | 44.40 | 81.72 | 73.09 | 67.57 |
| | AdaLoRA | 16 | 53.07 | 82.03 | 81.99 | 50.11 | 78.61 | 45.40 | 81.28 | 74.11 | 68.33 |
| | AdaLoRA | 8 | 53.33 | 82.03 | 82.11 | 49.13 | 78.57 | 45.20 | 81.34 | 73.79 | 68.19 |
| Rank = 16 | AdaLoRA | 4 | 50.85 | 80.72 | 80.73 | 37.98 | 77.34 | 42.80 | 80.52 | 73.16 | 65.51 |
| | LoftQ | $4^1$ | 55.12 | 82.58 | 82.69 | 49.81 | 78.82 | 45.80 | 81.28 | 74.27 | 68.80 |
| | LoftQ | $4^5$ | 53.92 | 82.32 | 81.56 | 42.00 | 78.54 | 43.80 | 81.56 | 72.77 | 67.06 |
| | LoftQ | $4^{10}$ | 52.90 | 81.69 | 81.56 | 39.88 | 78.64 | 43.80 | 81.07 | 71.98 | 66.44 |
| | QuaRot | 4 | 54.23 | 82.28 | 82.01 | 50.89 | 78.58 | 45.20 | 81.45 | 73.18 | 68.48 |
| | SpinQuant | 4 | 54.52 | 82.45 | 82.15 | 51.28 | 78.74 | 45.20 | 81.56 | 73.32 | 68.70 |
| | QR-Adaptor (≤4-bit) | 3.625 | 56.15 | 82.78 | 82.45 | 54.12 | 79.58 | 45.60 | 82.12 | 75.01 | 69.73 |
| | QR-Adaptor (Optimal) | 5.45 | 56.83 | **84.12** | **83.38** | **56.29** | **80.93** | 45.80 | **82.92** | **75.10** | **70.67** |

observed value $f(C^+)$, balancing exploration and exploitation:

$$\text{EI}(C^*) = (\mu(C^*) - f(C^+))\Phi(Z) + \sigma(C^*)\phi(Z)$$
$$\text{with } Z = \frac{\mu(C^*) - f(C^+)}{\sigma(C^*)}, \tag{7}$$

where $\Phi(\cdot)$ and $\phi(\cdot)$ are the CDF and PDF of the standard normal distribution. This sample-efficient process allows us to quickly converge on a refined optimal solution $C^*$ that best satisfies the user-defined preference $\alpha$. The visual flowchart is in Figure 4 in the Appendix

## 4 EVALUATION

In this section, we first introduce the experimental setup, including datasets, models, baselines, and implementation details. All hyperparameters aside from rank value and bit-width are kept consistent with the baselines.

### 4.1 EXPERIMENTAL SETUP

**Datasets and LLMs.** We utilize the Alpaca52k and hc3 (Taori et al., 2023) for fine-tuning and evaluate the zero-shot performance of these LLMs on benchmarks including BoolQ (Clark et al., 2019), PIQA (Bisk et al., 2020), HellaSwag (Zellers et al., 2019), WinoGrande (Sakaguchi et al., 2021), ARC-easy (Clark et al., 2018), ARC-challenge (Clark et al., 2018), OpenbookQA (Mihaylov et al., 2018), and MMLU Hendrycks et al. (2021). The models used in our experiments are LLaMA2 Touvron et al. (2023), LLaMA3.1 Grattafiori et al. (2024), LLaMA3.2, and Qwen2.5 Qwen et al. (2025). These models cover a range of scales and architectures to demonstrate the generalizability of our approach across different model families.

**Baselines.** We compare our method against several baselines: without tuning, LoRA Hu et al. (2022), QLoRA Dettmers et al. (2023), Adalora Zhang et al. (2023b), LoftQ Li et al. (2023), and

**Table 3:** Performance comparison across different model architectures (r=8). Bold figures represent the best performance for each model. Accuracy is reported as %.

| Model | Method | Bit | ARC(C) | ARC(E) | BoolQ | GSM8K | HellaS | OBQA | PIQA | WinoG | Average |
|---|---|---|---|---|---|---|---|---|---|---|---|
| Qwen-2.5-7B | LoRA | 16 | 56.01 | 83.48 | 82.97 | 54.03 | 79.01 | 45.00 | 81.95 | 74.98 | 69.68 |
| | QLoRA | 4 | 54.02 | 82.04 | 81.53 | 44.11 | 78.02 | 44.00 | 81.04 | 72.96 | 67.22 |
| | AdaLoRA | 4 | 51.03 | 80.51 | 80.04 | 37.23 | 77.04 | 42.60 | 80.53 | 72.01 | 65.11 |
| | LoftQ | 4[1] | 53.96 | 82.15 | 81.87 | 43.84 | 77.93 | 43.80 | 80.72 | 72.54 | 67.11 |
| | QR-Adaptor ($\leq$ 4bit) | 3.875 | 54.89 | 82.71 | 82.25 | 49.87 | 78.73 | 45.20 | 81.49 | 73.40 | 68.56 |
| | QR-Adaptor (Optimal) | 5.125 | **56.52** | **84.01** | **83.49** | **56.03** | **80.52** | **46.00** | **82.51** | **75.52** | **70.58** |
| Qwen-2.5-3B | LoRA | 16 | 52.98 | 81.03 | 80.01 | 45.02 | 76.01 | 42.00 | 79.03 | 70.99 | 65.88 |
| | QLoRA | 4 | 51.01 | 79.02 | 79.03 | 36.04 | 75.01 | 41.00 | 78.53 | 68.97 | 63.51 |
| | AdaLoRA | 4 | 49.03 | 78.01 | 78.02 | 29.01 | 74.03 | 40.00 | 77.01 | 68.03 | 61.64 |
| | LoftQ | 4[1] | 50.92 | 79.23 | 78.87 | 35.48 | 74.95 | 40.60 | 77.87 | 68.65 | 63.32 |
| | QR-Adaptor ($\leq$ 4bit) | 3.375 | 51.87 | 79.91 | 79.76 | 41.03 | 75.45 | 41.80 | 78.43 | 69.41 | 64.69 |
| | QR-Adaptor (Optimal) | 4.875 | **53.53** | **81.51** | **80.52** | **47.01** | **77.03** | **43.00** | **79.51** | **71.52** | **66.70** |
| LLaMA-3.2-3B | LoRA | 16 | 53.51 | 81.23 | 80.51 | 46.03 | 76.51 | 42.60 | 79.52 | 71.31 | 66.39 |
| | QLoRA | 4 | 51.52 | 79.51 | 79.52 | 37.01 | 75.53 | 41.60 | 78.53 | 69.51 | 64.08 |
| | AdaLoRA | 4 | 49.53 | 78.52 | 78.51 | 30.03 | 74.52 | 40.60 | 77.51 | 68.52 | 62.21 |
| | LoftQ | 4[1] | 51.78 | 79.83 | 79.87 | 37.42 | 75.78 | 41.20 | 78.72 | 69.84 | 64.49 |
| | QR-Adaptor ($\leq$ 4bit) | 3.75 | 52.41 | 80.25 | 80.17 | 42.01 | 75.95 | 42.20 | 78.96 | 69.95 | 65.23 |
| | QR-Adaptor (Optimal) | 5.375 | **54.01** | **81.83** | **81.02** | **48.01** | **77.52** | **43.60** | **80.01** | **72.03** | **67.24** |

LQ-LoRA Guo et al. (2024). We evaluated the performance of LoftQ with different iteration numbers. For Adalora, which dynamically allocates ranks based on the average rank budget, we set the budget to 8 and 64. Finally, for LQ-LoRA, which allocates quantization bit-width based on the average weight bit-width budget and quantization error, we set the bit-width budget to 4. Additionally, we include recent 4-bit quantization methods: QuaRot Ashkboos et al. (2024), which uses random rotations to handle outliers, and SpinQuant Liu et al. (2024b), which employs learned rotations for optimal quantization accuracy. For extreme low-bit comparison, we evaluate against 2-bit methods including ApiQ Liao et al. (2024) and RILQ Lee et al. (2025), both utilizing LoRA-based quantization error compensation.

**Implementation Details.** We utilize the following configurations: *PyTorch* version 2.1.2, *BitsandBytes* library version 0.43.1, *Transformers* library version 4.41.0, *PEFT* library version 0.11.1, *Optuna* library version 3.6.1, *CUDA* version 12.4, *GPU:* NVIDIA L20. *Operating System:* Ubuntu. Concise implementation details are provided in the Appendix I. We define the population size as 5 and generate 1 new offspring in each iteration. The second and third phases were iterated 5 times.

## 4.2 MAIN RESULTS

We present the performance comparison of LLaMA3.1-8B on commonsense understanding tasks in Table 2. We further evaluate QR-Adaptor on three additional models (Qwen-2.5-7B, Qwen-2.5-3B, and LLaMA-3.2-3B) as shown in Table 3. For practical deployment considerations, we include both $\leq$ 4-bit constrained and optimal configurations. Our method achieves or surpasses the performance of 16-bit fine-tuned models while maintaining competitive memory usage.

As mentioned, LoftQ outperforms 4-bit QLoRA after one iteration, but its performance degrades with more iterations due to the mismatch between continuous error correction and discrete quantization. Recent 4-bit methods like QuaRot and SpinQuant, which use rotation-based outlier handling, show competitive results but still fall short of our approach. For extreme quantization, we compare against 2-bit methods including ApiQ and RILQ. QR-Adaptor demonstrates consistent improvements across all bit-width regimes.

QR-Adaptor jointly optimizes bit-width and LoRA rank, balancing precision and adaptation capacity for superior performance. Notably, even under strict $\leq$ 4-bit constraints (average 3.625 bits), QR-Adaptor achieves 69.37% average accuracy, outperforming all 4-bit baselines. In the optimal configuration (5.45 bits average), it reaches 70.67%, substantially exceeding both traditional and recent quantization methods. Furthermore, QR-Adaptor allocates resources more efficiently by assigning higher LoRA ranks to critical layers and higher precision to important layers, achieving high accuracy with low memory usage.

Due to space constraints, additional experimental results and analyses are provided in Appendix G, including: (1) extended results across different models (G.1, G.2, G.5, G.6); (2) evaluations on larger datasets with higher LoRA ranks (G.3).

**Table 5:** Performance comparison of different methods across various bit-width configurations on Llama3.1-8B with higher ranks. Bold figures represent the best performance for a given model and task, while underlined figures indicate the second-best. QR-Adaptor* is transferred config. Accuracy is reported as %.

| Method | Rank | Bit | ARC(C) | ARC(E) | BoolQ | HellaS | OBQA | PIQA | WinoG | MMLU | Average |
|---|---|---|---|---|---|---|---|---|---|---|---|
| LoRA | 32 | 16 | 54.86 | 82.74 | 82.75 | _79.21_ | 44.40 | _81.99_ | 74.11 | 63.66 | 70.47 |
| LoRA | 64 | 16 | _55.46_ | 82.95 | _82.94_ | 79.13 | 45.00 | 81.88 | _74.51_ | _64.34_ | _70.78_ |
| QLoRA | 32 | 8 | 55.20 | _83.12_ | 81.93 | 79.07 | **46.20** | 81.88 | 73.32 | 63.28 | 70.50 |
| QLoRA | 32 | 4 | 53.41 | 80.89 | 82.05 | 78.42 | 43.60 | 80.90 | 73.01 | 60.97 | 69.16 |
| QLoRA | 64 | 8 | _55.46_ | 83.04 | 81.96 | 79.17 | _45.80_ | 81.94 | 73.01 | 63.34 | 70.47 |
| QLoRA | 64 | 4 | 53.41 | 81.19 | 81.74 | 78.35 | 44.60 | 80.69 | 72.06 | 60.79 | 69.10 |
| AdaLoRA | 32 | 8 | 53.92 | 81.82 | 82.20 | 78.57 | **46.20** | 81.50 | 73.40 | 63.82 | 70.18 |
| AdaLoRA | 32 | 4 | 51.45 | 81.02 | 80.86 | 77.30 | 42.40 | 80.96 | 72.53 | 58.15 | 68.08 |
| AdaLoRA | 64 | 8 | 53.92 | 82.11 | 81.93 | 78.74 | 46.20 | 81.39 | 73.95 | 63.88 | 70.27 |
| AdaLoRA | 64 | 4 | 52.13 | 80.98 | 81.04 | 77.20 | 42.20 | 80.85 | 72.77 | 58.07 | 68.16 |
| LoftQ | 32 | $4^1$ | 53.84 | 81.36 | 81.41 | 78.12 | 43.00 | 81.50 | 73.56 | 59.40 | 69.02 |
| LoftQ | 32 | $4^5$ | 52.56 | 81.36 | 81.96 | 78.05 | 42.80 | 81.45 | 73.09 | 59.41 | 68.84 |
| LoftQ | 32 | $4^{10}$ | 51.62 | 81.31 | 82.51 | 78.16 | 43.60 | 81.34 | 72.30 | 59.12 | 68.75 |
| LoftQ | 64 | $4^1$ | 52.82 | 81.40 | 81.59 | 78.23 | 43.20 | 81.34 | 73.88 | 59.78 | 69.03 |
| LoftQ | 64 | $4^5$ | 52.39 | 81.10 | 81.13 | 78.33 | 43.40 | 81.34 | 73.24 | 58.69 | 68.70 |
| LoftQ | 64 | $4^{10}$ | 51.71 | 81.23 | 81.62 | 78.37 | 43.20 | 81.01 | 72.77 | 59.25 | 68.65 |
| QR-Adaptor* | 32 | 3.625 | 55.23 | 82.89 | 82.65 | 79.12 | 45.40 | 81.77 | 73.88 | 63.78 | 70.59 |
| QR-Adaptor | 32 | 5.875 | **56.12** | **83.45** | **83.21** | **79.78** | **46.20** | **82.10** | **74.59** | **64.40** | **71.23** |

## 4.3 Computational Efficiency

The computational cost of QR-Adaptor primarily stems from actual performance testing on calibration data during stages 2 and 3. Specifically, after predicting a configuration, we need to conduct real performance tests to validate predictions. The prediction process itself is extremely fast (second-level), while one performance test on LLaMA3.1-8B model requires approximately 8-9 minutes. In comparison, one LoftQ iteration takes 11 minutes, and AdaLoRA, due to its dynamic rank adjustment during training, is typically 30-35% slower than LoRA.

QR-Adaptor's advantage becomes evident in resource-constrained scenarios where performance optimization is critical. While the initial search requires multiple performance evaluations, the method can continuously improve model performance through iterative optimization. Moreover, the three-stage design ensures efficient exploration of the configuration space, requiring fewer total evaluations compared to exhaustive search approaches.

To provide a comprehensive time comparison, we conducted experiments fine-tuning LLaMA3.1-8B on Alpaca52k dataset for 2 epochs across different methods. The results are summarized in Table 4.

To address potential concerns about computational overhead, we investigated the transferability of optimized configurations across datasets. We found that configurations optimized on one dataset exhibit significant transferability when applied to fine-tuning on different datasets. Specifically, when directly using a configuration optimized on dataset A for fine-tuning on dataset B, we still achieve notable performance improvements compared to baseline methods, albeit not as substantial as performing optimization specifically for dataset B.

As shown in Table 9, the "QR-Adaptor (Transferred Config)" row demonstrates the performance when directly applying a configuration optimized on a 52k dataset to the larger 177k dataset. This demonstrates that the same model's optimized configurations possess transferability, enabling direct use in fine-tuning on different datasets while still achieving significant performance improvements

**Table 4:** Training time comparison of different methods on LLaMA3.1-8B with Alpaca52k dataset (2 epochs). All experiments were conducted on L20 GPU.

| Method | Total Time (minutes) |
|---|---|
| LoRA | 300 |
| QLoRA | 360-405 |
| AdaLoRA | 390-405 |
| LoftQ (1 iteration) | 370-416 |
| LoftQ (5 iterations) | 415-460 |
| LoftQ (10 iterations) | 470-515 |
| QR-Adaptor | 445-495 |

### 4.4 ABLATION STUDY

We use the WinoGrande benchmark to conduct an ablation study assessing the contribution of each stage in QR-Adaptor. As shown in Figure 1, removing either PRGA or Bayesian optimization leads to unbalanced search behavior—PRGA alone explores too broadly, while Bayesian optimization alone is overly narrow—reflecting their extrapolation and interpolation roles, respectively. Omitting stage 1 causes PRGA to initiate from random configurations, resulting in scattered search patterns. Nonetheless, it still reaches the upper-left optimal region, highlighting the strength of PRGA and Bayesian optimization. In contrast, the full three-stage pipeline first explores broadly around a guided initialization, then refines near promising areas, yielding the best configurations.

Further ablation studies assess the impact of each stage by removing them individually and analyzing the resulting performance. We also perform sensitivity analysis on PRGA hyperparameters, with details provided in Appendix J.

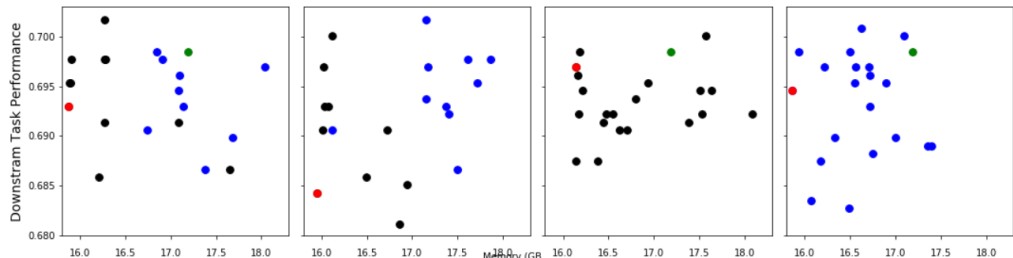

**Figure 1:** From left to right, the actual measured performance and memory usage of the configurations generated by QR-Adaptor, QR-Adaptor without stage1, QR-Adaptor without stage2, and QR-Adaptor without stage3 are shown. Different colors represent the configurations generated at different stages.

## 5 RELATED WORK

**LLM Quantization.** LLM quantization enables efficient deployment by reducing precision. Notable methods include GPTQ Frantar et al. (2023), AWQ Lin et al. (2023), SmoothQuant Xiao et al. (2023), ZeroQuant Yao et al. (2022), and LLM.int8() Dettmers et al. (2022). Mixed-precision approaches such as APTQ Guan et al. (2024), MixLLM Wang et al. (2025), and SliM-LLM Huang et al. highlight the importance of per-layer precision allocation, though they focus solely on quantization.

**Parameter Efficient Fine-Tuning.** PEFT methods enhance LLMs without heavy inference costs. QLoRA Dettmers et al. (2023) and LoftQ Li et al. (2023) combine quantization with low-rank adapters. Variants such as AdaLoRA Zhang et al. (2023a), LQ-LoRA Guo et al. (2023), RankAdaptor Zhou et al. (2024), and DoRA Liu et al. (2024a) emphasize adaptive allocation across layers, but remain independent of quantization.

**Joint Quantization and Low-Rank Adaptation.** LoftQ Li et al. (2023) alternates between quantization and low-rank approximation, while LQ-LoRA Guo et al. (2023) combines the two under memory constraints. However, existing approaches typically optimize quantization and adaptation separately, leaving joint allocation of precision and rank underexplored. More related work is in the Appendix A.

## 6 CONCLUSION

In this work, we propose QR-Adaptor, a unified, gradient-free method that uses partial calibration data to simultaneously optimize the precision and LoRA rank of each model layer. By focusing on the discrete nature of quantization and low-rank spaces and optimizing them within a task-driven framework, QR-Adaptor overcomes the limitations of iterative error-fitting techniques and rank-adaptive methods unsuitable for quantization. Our extensive experiments demonstrate that QR-Adaptor consistently outperforms existing baselines, achieving better performance than 16-bit fine-tuned models while maintaining a 4-bit memory footprint. These results highlight the importance of integrating quantization and low-rank matrices into a single, cohesive optimization process, driven by actual performance and memory efficiency.

## ETHICS STATEMENT

This work builds upon pre-trained large language models Llama2 and utilizes publicly available datasets for instruction fine-tuning Alpaca-clean. We do not introduce any new datasets or data collection processes, and therefore do not involve human annotation in this research. Additionally, our study focuses on improving model efficiency through pruning and quantization techniques, without engaging with sensitive content or user-specific data. As such, this paper does not present any ethical concerns beyond those already associated with the broader body of research on large language models and their datasets. All datasets and models used comply with their respective licenses and terms of use.

## REPRODUCIBILITY STATEMENT

To ensure the reproducibility of our results, we provide comprehensive documentation on the steps required to replicate our experiments. Our code is available in scripts such as `optuna_main-v3.py`, `post_training_mixed_quant.py`, and `run_optuna.py`, which handle hyperparameter optimization, mixed-precision quantization, and evaluation. For data preparation, we utilize the Alpaca Cleaned Dataset from `yahma/alpaca-cleaned`, which is automatically downloaded and processed using the `datasets` library. Our environment setup requires an NVIDIA GPU with CUDA support, preferably with at least 20 GB of memory for the Llama2 model, as well as Python 3.8+ and dependencies like PyTorch, Transformers, Optuna, BitsAndBytes, PEFT, and other libraries, which can be installed via the `requirements.txt` file. The model we fine-tune is the Llama2 architecture (`NousResearch/Llama-2-7b-hf`), using a mixed-precision quantization approach via `bitsandbytes` and Low-Rank Adaptation (LoRA) with the `peft` library. The training is conducted using a mixed-precision setup where the model's dtype is set to `torch.bfloat16` to optimize memory usage and computation efficiency. Our hyperparameter optimization framework leverages Optuna to maximize model accuracy while minimizing memory usage, tuning parameters like quantization bits (4 or 8 bits) and LoRA ranks (2 to 16). To replicate our training process, researchers can execute the provided scripts using the specified command-line arguments, which configure the model, output directories, number of trials, and evaluation tasks. Model checkpoints and Optuna results are saved at regular intervals. The training is conducted using the Hugging Face `Trainer`, configured with parameters including a batch size of 4, gradient accumulation steps of 16, warmup steps of 100, and a learning rate of 1e-4, with evaluation and model saving steps set to every 200 steps. Evaluation is conducted using the `lm_eval` library, where metrics such as accuracy are recorded and saved in JSON format. All hyperparameter settings and model configurations are logged in the output directory, along with training progress and memory usage. Random seeds are set to ensure deterministic behavior. By following these steps, including hardware and software specifications, and running the scripts with the provided configurations, researchers can reproduce our experiments and validate the findings related to mixed-precision quantization and parameter-efficient fine-tuning.

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

# Table of Contents

## A    EXTENDED RELATED WORK

**LLM Quantization.** The field of LLM quantization has witnessed substantial progress, driven by the need for efficient model deployment. Recent research has introduced several innovative approaches. Frantar et al. (2023) developed GPTQ, achieving 4-bit precision with layer-wise quantization. Lin et al. (2023) proposed AWQ, improving accuracy for heavily quantized models. Xiao et al. (2023) introduced SmoothQuant, addressing quantization of both weights and activations. Yao et al. (2022) introduced ZeroQuant, preserving zero-shot capabilities at low bit widths. Dettmers et al. (2022) presented LLM.int8(), enabling 8-bit quantization on consumer hardware. Kim et al. (2023) combined quantization with pruning and knowledge distillation in SqueezeLLM. Mixed-precision quantization has further advanced the field: APTQ Guan et al. (2024) balances compression and performance, MixLLM Wang et al. (2025) employs dynamic strategies, and SliM-LLM Huang et al. provides another mixed-precision solution. These works highlight the importance of per-layer allocation but focus exclusively on quantization.

**Parameter Efficient Fine-Tuning.** PEFT techniques enhance LLMs without raising inference costs. QLoRA Dettmers et al. (2023) combines 4-bit quantization with low-rank adapters, while LoftQ Li et al. (2023) alternates between quantization and low-rank steps. CoLoRA Berman & Peherstorfer (2024) accelerates predictions under new parameters. AdaLoRA Zhang et al. (2023a) adaptively allocates update budgets; LQ-LoRA Guo et al. (2023) merges decomposition and quantization; RankAdaptor Zhou et al. (2024) enables hierarchical dynamic adaptation; DoRA Liu et al. (2024a) decomposes weights into magnitude and direction. While these approaches account for heterogeneous adaptation needs, they largely remain orthogonal to quantization.

**Joint Quantization and Low-Rank Adaptation.** Some works integrate quantization and adaptation. LoftQ Li et al. (2023) alternates quantization with low-rank approximation using SVD initialization, but iterative error fitting may degrade performance. LQ-LoRA Guo et al. (2023) combines low-rank decomposition with quantization, allocating bit-widths based on error budgets, though rank is treated separately. These methods are important steps but optimize quantization and adaptation independently or via proxy metrics, leaving open the challenge of jointly allocating memory between precision and rank per layer.

**Neural Architecture Search and Optimization.** Joint optimization of quantization and adaptation parameters connects to neural architecture search. DARTS Liu et al. introduced differentiable architecture search, and HAT Wang et al. (2020) proposed hardware-aware transformers. However, they primarily explore architecture design, not the discrete per-layer precision–rank allocation problem. Our discrete optimization requires specialized search strategies; genetic algorithms and Bayesian optimization offer promising directions, motivating our three-stage approach combining task-informed initialization, global exploration, and local refinement.

## B    USE OF LLMS

In preparing this paper, LLMs were employed solely for language refinement purposes, such as improving grammar, clarity, and style of expression. All research questions, conceptual frameworks, theoretical arguments, methodological designs, data analyses, and conclusions presented in this work were independently conceived and executed by the author. The LLMs did not generate, alter, or influence the underlying ideas, interpretations, or findings. Their use was limited to assisting in polishing the readability and fluency of the manuscript while preserving the originality and integrity of the scholarly contributions.

## C    ON THE SUBOPTIMALITY OF CONTINUOUS PROXIES FOR DISCRETE QUANTIZATION

This appendix provides a formal analysis of why iterative fine-tuning methods that rely on continuous, low-rank updates to correct quantization error can be suboptimal. These methods, such as LoftQ, operate by minimizing a continuous objective (e.g., Frobenius norm) but ultimately must project the result back into a discrete space for inference. We demonstrate that this two-stage process does not

guarantee finding the optimal discrete representation, thereby motivating our gradient-free, direct search approach in the discrete configuration space.

## C.1 Problem Formulation: Quantization as Projection

We begin by defining the quantization process as a projection onto a discrete lattice.

**Definition C.1** (Quantization Lattice and Operator). For a given bit-width, the set of representable scalar values forms a uniform grid with step size $\Delta$. For a weight matrix $\mathbf{W} \in \mathbb{R}^{d \times k}$, the corresponding quantization lattice $\Lambda$ is the set of all matrices whose elements belong to this grid:

$$\Lambda = \{\mathbf{M} \in \mathbb{R}^{d \times k} \mid M_{ij} = n_{ij}\Delta \text{ for some } n_{ij} \in \mathbb{Z}\}. \tag{8}$$

The standard quantization operator, $\text{Quantize}(\cdot)$, performs an element-wise rounding operation that maps a continuous matrix to the nearest point in the lattice $\Lambda$. This operator is equivalent to a projection onto $\Lambda$:

$$\mathbf{W}_q = \text{Quantize}(\mathbf{W}) = \mathcal{P}_\Lambda(\mathbf{W}) = \arg\min_{\mathbf{M} \in \Lambda} \|\mathbf{W} - \mathbf{M}\|_F. \tag{9}$$

The core objective of quantization-aware fine-tuning is to find a matrix $\mathbf{W}'_q \in \Lambda$ that not only minimizes the memory footprint but also maximizes downstream task performance. Ideally, this $\mathbf{W}'_q$ should be a good approximation of the optimal full-precision weights $\mathbf{W}^*$ for a given task. For simplicity in this analysis, we consider the goal to be finding the closest lattice point to a target full-precision matrix $\mathbf{W}$, i.e., finding $\mathcal{P}_\Lambda(\mathbf{W})$.

## C.2 Analysis of Iterative Low-Rank Refinement

Methods like LoftQ attempt to improve upon the initial quantization $\mathbf{W}_q = \mathcal{P}_\Lambda(\mathbf{W})$ by adding a continuous low-rank correction. The process can be described as follows:

1. **Initial Quantization:** Start with the baseline quantized matrix $\mathbf{W}_q = \mathcal{P}_\Lambda(\mathbf{W})$.

2. **Continuous Error Correction:** Define the quantization error as $\mathbf{E} = \mathbf{W} - \mathbf{W}_q$. Find a low-rank approximation $\mathbf{AB}^*$ to this error by minimizing a continuous objective:

$$\mathbf{AB}^* = \arg\min_{\text{rank}(\mathbf{AB}) \leq r} \|\mathbf{E} - \mathbf{AB}\|_F = \arg\min_{\text{rank}(\mathbf{AB}) \leq r} \|\mathbf{W} - (\mathbf{W}_q + \mathbf{AB})\|_F. \tag{10}$$

   The solution is typically found via Singular Value Decomposition (SVD) of the error matrix $\mathbf{E}$.

3. **Final Discretization:** The resulting matrix, $\mathbf{W}_{\text{updated}} = \mathbf{W}_q + \mathbf{AB}^*$, is continuous and not in $\Lambda$. For inference, it must be re-quantized:

$$\mathbf{W}'_q = \mathcal{P}_\Lambda(\mathbf{W}_{\text{updated}}) = \mathcal{P}_\Lambda(\mathbf{W}_q + \mathbf{AB}^*). \tag{11}$$

The critical question is whether this process reliably yields the best possible discrete approximation. That is, does $\mathbf{W}'_q$ equal the true optimal solution, $\mathcal{P}_\Lambda(\mathbf{W})$?

## C.3 Formal Argument for Suboptimality

The procedure described above is suboptimal because the continuous optimization in Step 2 is disconnected from the final discrete projection in Step 3.

**Proposition C.2.** Let $\mathbf{W}_{opt} = \mathcal{P}_\Lambda(\mathbf{W})$ be the optimal discrete approximation of $\mathbf{W}$. Let $\mathbf{W}'_q = \mathcal{P}_\Lambda(\mathbf{W}_q + \mathbf{AB}^*)$ be the matrix obtained from the iterative refinement process. It is not guaranteed that $\mathbf{W}'_q = \mathbf{W}_{opt}$. In high-dimensional spaces, they are often different.

*Justification.* The projection operator $\mathcal{P}_\Lambda$ partitions the continuous space $\mathbb{R}^{d \times k}$ into a set of disjoint Voronoi cells, one for each lattice point $\mathbf{M} \in \Lambda$. A point $\mathbf{Y}$ is projected to $\mathbf{M}$ if and only if it lies within the Voronoi cell of $\mathbf{M}$, denoted $V(\mathbf{M})$.

1. By definition, the optimal discrete solution is $\mathbf{W}_{\text{opt}} = \mathcal{P}_\Lambda(\mathbf{W})$, which means the original matrix $\mathbf{W}$ lies inside the Voronoi cell $V(\mathbf{W}_{\text{opt}})$. In our simplified case where $\mathbf{W}_q$ is the initial rounding, $\mathbf{W}_{\text{opt}} = \mathbf{W}_q$, so $\mathbf{W} \in V(\mathbf{W}_q)$.

2. The iterative method computes an updated continuous matrix $\mathbf{W}_{\text{updated}} = \mathbf{W}_q + \mathbf{A}\mathbf{B}^*$. Substituting the definitions, we have:

$$\mathbf{W}_{\text{updated}} = \mathbf{W} - \mathbf{E} + \mathbf{A}\mathbf{B}^* = \mathbf{W} - (\mathbf{E} - \mathbf{A}\mathbf{B}^*). \tag{12}$$

Here, $\mathbf{E} - \mathbf{A}\mathbf{B}^*$ is the residual error from the low-rank approximation of the quantization error $\mathbf{E}$.

3. The final quantized matrix is $\mathbf{W}_q' = \mathcal{P}_\Lambda(\mathbf{W}_{\text{updated}})$. This means $\mathbf{W}_{\text{updated}}$ must lie in the Voronoi cell $V(\mathbf{W}_q')$.

4. For the method to be optimal (i.e., $\mathbf{W}_q' = \mathbf{W}_{\text{opt}}$), the updated point $\mathbf{W}_{\text{updated}}$ must lie in the same Voronoi cell as the original point $\mathbf{W}$. However, the term $\delta = \mathbf{E} - \mathbf{A}\mathbf{B}^*$ acts as a perturbation on $\mathbf{W}$. The quantization error $\mathbf{E}$ is typically a dense, noisy, high-rank matrix. Its low-rank approximation error $\delta$ is therefore also a high-rank matrix.

Adding this high-rank perturbation $\delta$ to $\mathbf{W}$ can easily push the vector across a Voronoi boundary into an adjacent cell. When $\mathbf{W} - \delta$ falls into a different cell $V(\mathbf{M})$ where $\mathbf{M} \neq \mathbf{W}_{\text{opt}}$, the final projection becomes suboptimal: $\mathcal{P}_\Lambda(\mathbf{W}_{\text{updated}}) = \mathbf{M} \neq \mathbf{W}_{\text{opt}}$. This occurs generically in high dimensions, as even a small perturbation has many dimensions in which it can push the vector across a boundary. The assumption that minimizing the continuous error $\|\mathbf{W} - \mathbf{W}_{\text{updated}}\|_F$ will keep $\mathbf{W}_{\text{updated}}$ in the correct Voronoi cell is unfounded. ∎

### C.4 THE DISCONTINUOUS OPTIMIZATION LANDSCAPE

The suboptimality issue is further compounded by the nature of the true underlying objective function. If we consider the downstream loss $\mathcal{L}$, the function we implicitly want to optimize with respect to $\mathbf{A}$ and $\mathbf{B}$ is:

$$f(\mathbf{A}, \mathbf{B}) = \mathcal{L}\left(\mathcal{P}_\Lambda(\mathbf{W}_q + \mathbf{A}\mathbf{B})\right). \tag{13}$$

Due to the discrete nature of the projection $\mathcal{P}_\Lambda$, this function is piecewise constant. Small, continuous changes to $\mathbf{A}$ and $\mathbf{B}$ will not change the output of $\mathcal{P}_\Lambda(\mathbf{W}_q + \mathbf{A}\mathbf{B})$ as long as the matrix remains within its current Voronoi cell. The function value only changes when $\mathbf{W}_q + \mathbf{A}\mathbf{B}$ crosses a boundary into a new cell.

This has a critical implication for optimization:

- **Vanishing Gradients:** The gradient $\nabla_{\mathbf{A}, \mathbf{B}} f(\mathbf{A}, \mathbf{B})$ is zero almost everywhere (within the interior of each Voronoi cell).

- **Optimization Stagnation:** Gradient-based methods are ineffective in such a landscape. They receive no signal to guide the updates of $\mathbf{A}$ and $\mathbf{B}$ and will stagnate unless an update is large enough to jump to a new discrete state.

**Conclusion.** The analysis reveals two fundamental flaws in using continuous proxies for discrete optimization. First, minimizing a continuous error metric does not guarantee finding the optimal discrete solution after re-quantization. Second, the true objective landscape is piecewise constant, rendering standard gradient-based optimization ineffective. These limitations strongly motivate a shift away from continuous proxies and towards methods that directly search the discrete space of configurations. Our QR-Adaptor framework, which uses a gradient-free, multi-objective search to evaluate discrete (bit-width, rank) configurations based on their actual downstream performance, is a principled response to these challenges.

## D  QUANTIZATION

We first apply NF-quantization with bit size $b_0$ and bucket size $B_0$ to obtain the quantized matrix $\widehat{A_i}$ and the absmax values for each block $s = [s_1, \ldots, s_{\frac{\text{sizeof}(A_i)}{B_0}}]$. These absmax values are further

quantized to $b_1$ bits via uniform integer quantization with bucket size $B_1$ to obtain the quantized vector $\widehat{s}$, along with the absmax values for $s$, i.e., $v = [v_1, \ldots v_{\frac{\text{sizeof}(A_i)}{B_0 B_1}}]$. Finally, we cast $v$ to $b_2$ bits to obtain $\widehat{v}$.

This quantization scheme requires storing $\widehat{A_i}, \widehat{s}, \widehat{v}$ to represent $A_i$. We can thus quantify the memory cost (number of bits) for storing $A_i$ given a configuration $c_i = (b_0, b_1, b_2, B_0, B_1)$ as:

$$\text{memory\_cost}(A_i, c_i) = \text{sizeof}(A_i) \cdot \left( b_0 + \frac{b_1}{B_0} + \frac{b_2}{B_0 \cdot B_1} \right) \quad (14)$$

The original NF-4 double quantization is a special case with $q_{\text{NF4}} = (4, 8, \text{fp32}, 64, 256)$ and $\text{memory\_cost}(A_i, q_{\text{NF4}}) = 4.127 \cdot \text{sizeof}(A_i)$, i.e., NF-4 requires on average 4.127 bits per parameter.

# E  QR-ADAPTOR SEARCH PROCESS DETAILS

This appendix provides supplementary details regarding the QR-Adaptor search methodology and its associated computational costs, addressing reproducibility and practical implementation concerns.

## E.1  SEARCH HYPERPARAMETERS AND CONFIGURATION

To ensure the reproducibility of our results, we list the specific hyperparameters and configurations used for the QR-Adaptor search process in Table 6. These settings were kept consistent across all main experiments unless otherwise noted.

**Table 6:** Hyperparameters for the QR-Adaptor search process.

| Parameter | Stage | Value / Description |
|---|---|---|
| **General Search Configuration** | | |
| Bit-width Search Space ($\mathcal{Q}$) | All | $\{2, 4, 8\}$ |
| LoRA Rank Search Space ($\mathcal{R}$) | All | $\{0, 2, 4, 6, 8, \ldots, 64\}$ |
| Calibration Dataset | All | A random subset of 1024 samples from the C4 dataset. |
| Fine-tuning Epochs (per evaluation) | All | 1 epoch on the calibration dataset. |
| **Stage 1: Task-Informed Initialization** | | |
| Importance Score Metric ($I(l)$) | Initialization | Gradient-based saliency score (magnitude of Fisher Information). |
| Initial Population Size ($N_{\text{pop}}$) | Initialization | 1 |
| **Stage 2: Global Exploration (PRGA)** | | |
| Algorithm | PRGA | NSGA-II (Non-dominated Sorting Genetic Algorithm II) |
| Number of Generations | PRGA | 5 |
| Population Size | PRGA | 20 |
| Selection Mechanism | PRGA | Tournament selection based on non-dominated rank and crowding distance. |
| Crossover Operator | PRGA | Uniform Crossover with a probability of 0.9. |
| Mutation Operator | PRGA | Per-layer random mutation: for each layer, with probability 0.1, re-sample its bit-width and rank from $\mathcal{Q}$ and $\mathcal{R}$. |
| **Stage 3: Local Refinement (Bayesian Optimization)** | | |
| Surrogate Model | BO | Gaussian Process (GP) |
| GP Kernel | BO | Matérn 5/2 kernel with Automatic Relevance Determination (ARD). |
| Acquisition Function | BO | Expected Improvement (EI). |
| Number of Iterations | BO | 5 iterations per configuration refined from the Pareto front. |

## E.2  TASK-INFORMED INITIALIZATION ALGORITHM

As mentioned in Section 3.2.1, the initialization process uses layer importance scores to generate a high-quality initial configuration. Algorithm 3 provides a concrete step-by-step description of this procedure. The core idea is to map higher importance scores to a higher probability of allocating more resources (i.e., higher bit-widths and ranks). **This single seed configuration $C_0$ is evaluated by fine-tuning for one epoch on the calibration dataset to measure its initial performance**, forming the starting point for the global search. The subsequent PRGA stage will generate a full population of size 20 through mutations and crossover operations based on this seed.

---

**Algorithm 3** Task-Informed Initialization Process

---

1: **Input:** Layer importance scores $\{I(l)\}_{l=1}^{L}$, Bit-width space $\mathcal{Q}$, Rank space $\mathcal{R}$.
2: **Output:** Seed configuration $C_0$.
3:              ▷ Step 1: Normalize importance scores to create a sampling distribution
4: Normalize scores: $p_l \leftarrow I(l)/\sum_{j=1}^{L} I(j)$ for $l = 1, \ldots, L$.
5:                   ▷ Step 2: Generate the seed configuration $C_0$ based on importance
6: Initialize $C_0 = [(\text{bit}_1, \text{rank}_1), \ldots, (\text{bit}_L, \text{rank}_L)]$.
7: **for** $l = 1$ to $L$ **do**
8:      // Map normalized importance $p_l$ to the search spaces.
9:      // The higher the importance, the higher the index in the sorted space.
10:      Sort $\mathcal{Q}$ and $\mathcal{R}$ in ascending order.
11:      Bit index $idx_b \leftarrow \lfloor p_l \cdot (|\mathcal{Q}| - 1) \rfloor$. Clamp to $[0, |\mathcal{Q}| - 1]$.
12:      Rank index $idx_r \leftarrow \lfloor p_l \cdot (|\mathcal{R}| - 1) \rfloor$. Clamp to $[0, |\mathcal{R}| - 1]$.
13:      $\text{bit}_l \leftarrow \mathcal{Q}[idx_b]$; $\text{rank}_l \leftarrow \mathcal{R}[idx_r]$.
14: **end for**
15:            ▷ Step 3: (Optional) Apply budget constraints if a target budget is predefined
16: **return** $C_0$.

---

# F   PSEUDO CODE OF THE SPECIFIC ALGORITHM IN THE QR-ADAPTOR FRAMEWORK

Due to page limitations, we present the pseudocode of the algorithm.

---

**Algorithm 4** Pareto Rank Calculation

---

1: **Input:** Population $P$ with $n$ individuals
2: Calculate the number of dominated individuals $n_p$ and the set of solutions dominated $S_p$ for each individual $p$
3: Place individuals with $n_p = 0$ into set $F_1$
4: **for** each individual $i$ in $F_1$ **do**
5:      **for** each individual $j \in S_i$ **do**
6:          $n_j \leftarrow n_j - 1$
7:          **if** $n_j = 0$ **then**
8:              Add individual $j$ to set $F_2$
9:          **end if**
10:      **end for**
11: **end for**
12: Repeat for $F_2, F_3, \ldots$, until all individuals are ranked
13: **Output:** Pareto-ranked individuals

---

**Algorithm 5** Crowding Distance Calculation

---

1: **Input:** Ranked individuals $F$ with $N$ individuals, $M$ objectives
2: **for** each individual $n \in 1 \ldots N$ **do**
3:      Initialize $d_n \leftarrow 0$
4: **end for**
5: **for** each objective function $f_m$ **do**
6:      Sort individuals based on $f_m$
7:      $f_m^{max}, f_m^{min} \leftarrow \max f_m, \min f_m$
8:      $d_1, d_N \leftarrow \infty$
9:      **for** $n = 2$ to $N - 1$ **do**
10:          $d_n \leftarrow d_n + \frac{f_m(n+1) - f_m(n-1)}{f_m^{max} - f_m^{min}}$
11:      **end for**
12: **end for**
13: **Output:** Crowding distances $d_n$ for each individual $n$

---

# G   MORE RESULTS

Due to page limitations, we present remaining results across various models here.

## G.1   EXPERIMENT SCOPE EXPANSION: LLAMA 2 SERIES

In the original experiments, the focus was primarily on Llama3.1, considering that its updated architecture present new challenges for quantization. Compared to Llama2 series, Llama3.1 is significantly harder to quantize, especially under low-bit configurations, as they incorporate more sophisticated architectural features. Additionally, to comprehensively demonstrate the superiority of **QR-Adaptor**, we have also conducted extensive performance experiments on the Llama2 series models, with the results presented in Table 7 and Table 8.

**Algorithm 6** Simulated Binary Crossover (SBX)

**Require:** Two parent individuals $P_1$ and $P_2$, each with $L$ real-valued genes
1: Initialize offspring $O_1$ and $O_2$ as empty
2: **for** $l = 1$ to $L$ **do**
3:     Generate a random number $u \in [0, 1]$
4:     **if** $u \leq 0.5$ **then**
5:         $\beta \leftarrow (2u)^{1/(n+1)}$
6:     **else**
7:         $\beta \leftarrow \left(\frac{1}{2(1-u)}\right)^{1/(n+1)}$
8:     **end if**
9:     $y_{1l} \leftarrow 0.5 \cdot ((1+\beta) \cdot p_{1l} + (1-\beta) \cdot p_{2l})$
10:    $y_{2l} \leftarrow 0.5 \cdot ((1-\beta) \cdot p_{1l} + (1+\beta) \cdot p_{2l})$
11:    Append $y_{1l}$ to $O_1$ and $y_{2l}$ to $O_2$
12: **end for**
13: **Output:** $O_1$ and $O_2$

**Algorithm 7** Polynomial Mutation

**Require:** Individual $P$ with $L$ real-valued genes, mutation probability $p_m$
1: Initialize mutated individual $P'$ as a copy of $P$
2: **for** $l = 1$ to $L$ **do**
3:     Generate a random number $u \in [0, 1]$
4:     **if** $u < p_m$ **then**
5:         Generate a random number $\delta \in [-1, 1]$
6:         $x'_l \leftarrow x_l + (x_{max} - x_{min}) \cdot \delta \cdot (1 - |\delta|)^{n-1}$
7:         Replace $x_l$ with $x'_l$ in $P'$
8:     **end if**
9: **end for**
10: **Output:** Mutated individual $P'$

**Table 7:** Performance comparison of different methods across various bit-width configurations on Llama2-7B. Superscripts on LoftQ bits indicate the number of initialization iterations. QR-Adaptor searches for optimal bit number and rank value for each layer based on different tasks with its bit number averaged across tasks. Bold figures represent the best performance for a given model and task, while underlined figures indicate the second-best. Accuracy is reported as %.

| | Method | Bit | ARC(C) | ARC(E) | BoolQ | HellaS | OBQA | PIQA | WinoG | Average |
|---|---|---|---|---|---|---|---|---|---|---|
| Rank = 8 | LoRA | 16 | 46.93 | 77.36 | 78.47 | **76.93** | 44.80 | 79.38 | 69.38 | 67.61 |
| | QLoRA | 8 | **48.21** | 77.36 | 77.92 | 76.88 | 44.80 | 79.82 | 68.75 | 67.70 |
| | QLoRA | 4 | 46.25 | 76.26 | 77.43 | 76.42 | **46.20** | 78.67 | 69.85 | 67.30 |
| | AdaLoRA | 16 | 46.08 | 76.77 | 77.46 | 75.89 | 44.20 | 79.16 | 69.22 | 66.97 |
| | AdaLoRA | 8 | 46.08 | 76.73 | 77.49 | 75.93 | 44.20 | 79.00 | 69.06 | 66.93 |
| | AdaLoRA | 4 | 46.33 | 75.25 | 76.39 | 75.45 | 44.40 | 77.91 | 69.14 | 66.41 |
| | LoftQ | $4^1$ | 46.16 | 77.10 | 77.43 | 76.68 | 44.80 | 79.33 | 69.30 | 67.26 |
| | LoftQ | $4^5$ | 47.35 | 76.64 | 76.33 | 76.36 | 45.60 | 79.05 | 69.06 | 67.20 |
| | LQ-LoRA | 4 | 47.18 | 76.60 | 76.54 | 76.24 | 45.00 | 78.84 | 68.90 | 67.04 |
| | QR-Adaptor | 5.45 | 48.04 | **77.44** | **78.96** | 76.84 | 46.00 | **79.86** | **69.97** | **68.15** |
| Rank = 16 | LoRA | 16 | 46.93 | **77.57** | 78.41 | 76.81 | 45.00 | 79.38 | 69.06 | 67.59 |
| | QLoRA | 8 | 47.61 | 77.44 | 78.41 | **76.93** | 45.40 | 79.05 | 69.06 | 67.70 |
| | QLoRA | 4 | 46.67 | 76.35 | 77.25 | 76.40 | 45.00 | 78.84 | 70.01 | 67.22 |
| | AdaLoRA | 16 | 46.16 | 76.68 | 77.58 | 75.92 | 44.20 | 79.11 | 69.38 | 67.00 |
| | AdaLoRA | 8 | 46.16 | 76.68 | 77.40 | 75.91 | 44.40 | 79.11 | 69.06 | 66.96 |
| | AdaLoRA | 4 | 46.33 | 75.29 | 76.45 | 75.44 | 44.20 | 77.91 | 69.46 | 66.47 |
| | LoftQ | $4^1$ | 47.10 | 77.19 | 77.89 | 76.61 | 44.80 | 79.43 | 69.69 | 67.53 |
| | LoftQ | $4^5$ | 47.95 | 76.47 | 76.79 | 76.25 | 45.60 | 78.51 | 69.61 | 67.31 |
| | LQ-LoRA | 4 | 47.10 | 76.39 | 77.22 | 76.33 | **46.40** | 78.78 | **70.09** | 67.47 |
| | QR-Adaptor | 5.45 | **48.04** | 77.44 | **78.96** | 76.84 | 46.00 | **79.86** | 69.97 | **68.15** |

Our results show that **QR-Adaptor** consistently demonstrates superior performance across all tasks and outperforms existing methods, such as AdaLoRA and LoftQ, on Llama 2 series. The robustness of **QR-Adaptor** is also evident, especially on tasks that typically cause performance degradation for other methods.

### G.2 VISUALIZATION RESULTS FOR THE MMLU TASK

The results for the MMLU task in LLaMA2 are shown in Figure 2. **QR-Adaptor** demonstrates outstanding performance across various benchmarks. Due to the rank value selection ranging from 2 to 16, in some cases, **QR-Adaptor** consumes less memory than the fine-tuned 4-bit quantized models. Moreover, the low-precision models fine-tuned by **QR-Adaptor** outperform the fine-tuned 16-bit

**Table 8:** Performance comparison of different methods across various bit-width configurations on Llama2-13B. Superscripts on LoftQ bits indicate the number of initialization iterations. QR-Adaptor searches for optimal bit number and rank value for each layer based on different tasks with its bit number averaged across tasks. Bold figures represent the best performance for a given model and task, while underlined figures indicate the second-best. Accuracy is reported as %.

|  | Method | Bit | ARC(C) | ARC(E) | BoolQ | HellaS | OBQA | PIQA | WinoG | Average |
|---|---|---|---|---|---|---|---|---|---|---|
| Rank = 8 | LoRA | 16 | 52.56 | 80.18 | 81.44 | 79.98 | **46.40** | 81.12 | 71.98 | 70.52 |
|  | QLoRA | 8 | 52.39 | 80.18 | 81.22 | 79.92 | 45.00 | 80.47 | **73.09** | 70.32 |
|  | QLoRA | 4 | 51.54 | 78.91 | 81.41 | 79.46 | 45.40 | 80.30 | 71.82 | 69.83 |
|  | AdaLoRA | 16 | 49.15 | 79.46 | 80.37 | 79.25 | 45.40 | 80.47 | 72.30 | 69.49 |
|  | AdaLoRA | 8 | 49.32 | 79.34 | 80.43 | 79.29 | 45.60 | 80.47 | 72.22 | 69.52 |
|  | AdaLoRA | 4 | 48.29 | 77.78 | 80.40 | 78.12 | 44.20 | 80.14 | 71.74 | 68.67 |
|  | LoftQ | $4^1$ | 50.68 | 78.79 | 81.16 | 79.12 | 45.80 | 80.41 | 71.35 | 69.62 |
|  | LoftQ | $4^5$ | 50.34 | 78.87 | 80.24 | 78.81 | 45.20 | 80.25 | 70.80 | 69.22 |
|  | LQ-LoRA | 4 | 50.60 | 78.79 | 80.67 | 78.91 | 45.00 | 80.14 | 71.11 | 69.32 |
|  | QR-Adaptor | 6.125 | **52.82** | **80.64** | **81.84** | **80.08** | 45.80 | **81.45** | 72.69 | **70.76** |
| Rank = 16 | LoRA | 16 | 52.13 | 79.84 | 81.50 | 80.07 | **46.20** | 81.23 | 71.98 | 70.42 |
|  | QLoRA | 8 | 51.54 | 80.01 | 81.13 | 79.86 | **46.20** | 81.18 | 72.22 | 70.31 |
|  | QLoRA | 4 | 51.45 | 79.04 | 81.04 | 79.48 | 45.60 | 80.47 | 71.82 | 69.84 |
|  | AdaLoRA | 16 | 49.40 | 79.34 | 80.46 | 79.28 | 45.40 | 80.47 | 72.30 | 69.52 |
|  | AdaLoRA | 8 | 49.49 | 79.29 | 80.40 | 79.27 | 45.40 | 80.52 | 72.38 | 69.54 |
|  | AdaLoRA | 4 | 48.29 | 77.69 | 80.43 | 78.10 | 44.20 | 80.09 | 71.67 | 68.64 |
|  | LoftQ | $4^1$ | 50.68 | 78.87 | 80.86 | 79.18 | 45.80 | 80.30 | 71.90 | 69.66 |
|  | LoftQ | $4^5$ | 50.60 | 78.96 | 80.92 | 79.15 | 45.40 | 80.41 | 71.59 | 69.58 |
|  | LQ-LoRA | 4 | 50.09 | 78.79 | 80.43 | 79.06 | 45.40 | 80.14 | 71.67 | 69.37 |
|  | QR-Adaptor | 6.125 | **52.82** | **80.64** | **81.84** | **80.08** | 45.80 | **81.45** | **72.69** | **70.76** |

models. Another advantage of the **QR-Adaptor** is that it can be implemented without any additional technical measures to optimize performance, apart from spending some time (about 15 minutes to get one data point). This simple but effective method is very useful in practical applications.

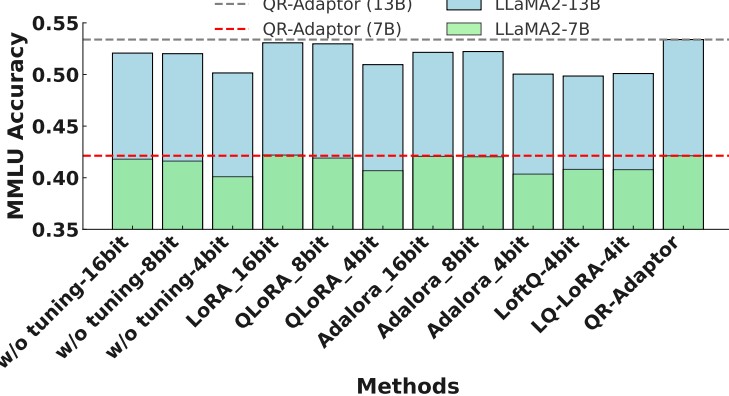

**Figure 2:** Performance comparison on MMLU benchmark. QR-Adaptor outperforms other methods.

### G.3 EFFECTIVENESS ON LARGER DATASETS WITH HIGHER RANKS

To address the concern regarding the effectiveness of small LoRA ranks on larger datasets, we conducted additional experiments on the LLaMA3.1-8B model using a larger dataset consisting of 177k samples. We tested our method with higher LoRA ranks (32 and 64) to evaluate its performance in handling large-scale data.

Our results are summarized in Table 9. The table compares the performance of **QR-Adaptor** with other baseline methods, including LoRA, QLoRA, AdaLoRA, and LoftQ, across various tasks. The

**Table 9:** Performance comparison of different methods across various bit-width configurations on Llama3.1-8B with higher ranks. Bold figures represent the best performance for a given model and task, while underlined figures indicate the second-best. QR-Adaptor* is transferred config. Accuracy is reported as %.

| Method | Rank | Bit | ARC(C) | ARC(E) | BoolQ | HellaS | OBQA | PIQA | WinoG | MMLU | Average |
|---|---|---|---|---|---|---|---|---|---|---|---|
| LoRA | 32 | 16 | 54.86 | 82.74 | 82.75 | 79.21 | 44.40 | 81.99 | 74.11 | 63.66 | 70.47 |
| LoRA | 64 | 16 | 55.46 | 82.95 | 82.94 | 79.13 | 45.00 | 81.88 | 74.51 | 64.34 | 70.78 |
| QLoRA | 32 | 8 | 55.20 | 83.12 | 81.93 | 79.07 | 46.20 | 81.88 | 73.32 | 63.28 | 70.50 |
| QLoRA | 32 | 4 | 53.41 | 80.89 | 82.05 | 78.42 | 43.60 | 80.90 | 73.01 | 60.97 | 69.16 |
| QLoRA | 64 | 8 | 55.46 | 83.04 | 81.96 | 79.17 | 45.80 | 81.94 | 73.01 | 63.34 | 70.47 |
| QLoRA | 64 | 4 | 53.41 | 81.19 | 81.74 | 78.35 | 44.60 | 80.69 | 72.06 | 60.79 | 69.10 |
| AdaLoRA | 32 | 8 | 53.92 | 81.82 | 82.20 | 78.57 | 46.20 | 81.50 | 73.40 | 63.82 | 70.18 |
| AdaLoRA | 32 | 4 | 51.45 | 81.02 | 80.86 | 77.30 | 42.40 | 80.96 | 72.53 | 58.15 | 68.08 |
| AdaLoRA | 64 | 8 | 53.92 | 82.11 | 81.93 | 78.74 | 46.20 | 81.39 | 73.95 | 63.88 | 70.27 |
| AdaLoRA | 64 | 4 | 52.13 | 80.98 | 81.04 | 77.20 | 42.20 | 80.85 | 72.77 | 58.07 | 68.16 |
| LoftQ | 32 | $4^1$ | 53.84 | 81.36 | 81.59 | 78.12 | 43.00 | 81.50 | 73.56 | 59.40 | 69.02 |
| LoftQ | 32 | $4^5$ | 52.56 | 81.36 | 81.96 | 78.05 | 42.80 | 81.45 | 73.09 | 59.41 | 68.84 |
| LoftQ | 32 | $4^{10}$ | 51.62 | 81.31 | 82.51 | 78.16 | 43.60 | 81.34 | 72.30 | 59.12 | 68.75 |
| LoftQ | 64 | $4^1$ | 52.82 | 81.40 | 81.59 | 78.23 | 43.20 | 81.34 | 73.88 | 59.78 | 69.03 |
| LoftQ | 64 | $4^5$ | 52.39 | 81.10 | 81.13 | 78.33 | 43.40 | 81.34 | 73.24 | 58.69 | 68.70 |
| LoftQ | 64 | $4^{10}$ | 51.71 | 81.23 | 81.62 | 78.37 | 43.20 | 81.01 | 72.77 | 59.25 | 68.65 |
| QR-Adaptor* | 32 | 3.625 | 55.23 | 82.89 | 82.65 | 79.12 | 45.40 | 81.77 | 73.88 | 63.78 | 70.59 |
| QR-Adaptor | 32 | 5.875 | **56.12** | **83.45** | **83.21** | **79.78** | **46.20** | **82.10** | **74.59** | **64.40** | **71.23** |

performance metrics include accuracy scores on datasets such as ARC (Challenge), ARC (Easy), BoolQ, HellaSwag, OpenBookQA, PIQA, WinoGrande, and MMLU.

KEY OBSERVATIONS

- **Effectiveness of LoRA Initialization**: Despite using higher ranks (32 and 64) and larger datasets, methods like LoftQ and LQ-LoRA do not consistently outperform the standard QLoRA baseline or the quantized models without fine-tuning. Increasing iterations in LoftQ (from LoftQ-1 to LoftQ-10) to better fit quantization errors leads to performance degradation, especially on challenging tasks like MMLU and GSM8K. These results suggest that fitting quantization errors using LoRA initialization is not universally effective and may introduce noise that hinders model performance.

- **Effectiveness on Larger Datasets**: Our method, **QR-Adaptor**, consistently achieves superior performance across all tasks and outperforms other methods, confirming its robustness and scalability. The results validate that **QR-Adaptor** is effective even when small LoRA ranks might not suffice for larger datasets.

- **Impact of Adaptive LoRA Rank Reduction**: AdaLoRA exhibits performance drops, particularly with lower bit-widths and on more challenging tasks. This supports our observation that dynamically adjusting the rank during fine-tuning can lead to convergence issues in quantized models, which are less robust due to quantization errors.

These results reinforce our initial observations and highlight the limitations of methods that attempt to fit quantization errors through LoRA initialization. The inability of LoftQ and AdaLoRA to improve performance significantly, even with higher ranks and larger datasets, underscores the challenges associated with such approaches. In contrast, **QR-Adaptor**, guided by our proposed constraints, demonstrates consistent performance improvements.

G.4 FAIRER COMPARISON: MATCHING BIT-WIDTH CONFIGURATIONS

Another important consideration for a fair comparison of quantization methods is the bit-width configuration used. To ensure that prior methods are evaluated under the same conditions as QR-Adaptor, we have re-evaluated AdaLoRA and LoftQ using the same mixed-precision configurations that were optimized through QR-Adaptor's framework. The updated results for Llama 2-13B are shown in Table 10.

The results indicate that the initialization constraints applied by QR-Adaptor provide substantial improvements over the original configurations of AdaLoRA and LoftQ. Despite these improvements, QR-Adaptor still outperforms these methods in terms of overall task performance. The constraints,

**Table 10:** Performance comparison with fair bit-width configurations for Llama2-13B. Accuracy is reported as %

| Method | BoolQ | PIQA | HellaS | WinoG | ARC(E) | ARC(C) | OBQA | Average |
|---|---|---|---|---|---|---|---|---|
| AdaLoRA | 81.08 | 80.13 | 79.21 | 71.74 | 79.51 | 50.12 | 45.60 | 69.77 |
| LoftQ | 80.93 | 79.47 | 79.02 | 71.34 | 79.26 | 51.20 | 45.60 | 69.98 |
| QR-Adaptor | **81.84** | **81.45** | **80.08** | **72.69** | **80.64** | **52.82** | **45.80** | **70.76** |

specifically ensuring stable initialization and fixing trainable parameters, contribute significantly to the enhanced performance of QR-Adaptor.

### G.5 Impact of Longer Fine-tuning Epochs on Unfixed Parameters

While increasing the fine-tuning epochs for AdaLoRA can lead to some performance improvements, these gains are marginal and AdaLoRA still does not outperform other methods like LoRA, QLoRA, or our proposed QR-Adaptor.

#### Findings

- **Marginal Improvement with Increased Epochs**: Extending the training of AdaLoRA from 2 epochs to 5 epochs results in a slight performance increase. However, this improvement is not substantial and comes at the cost of significantly longer training times.

- **Need for Mixed-Precision with Adaptive Rank**: The results suggest that adaptive rank adjustment alone, as in AdaLoRA, may not be the most effective approach. The combination of adaptive rank with mixed-precision quantization, as in QR-Adaptor, yields superior performance.

#### Supporting Data

We provide an updated table below that includes an "Epochs" column, showing the results for LoRA, QLoRA, AdaLoRA (at 2 and 5 epochs), and QR-Adaptor.

**Table 11:** Performance comparison of different methods with varying fine-tuning epochs on Llama3.1-8B. Accuracy is reported as %

| Method | Rank | Bit-width | Epochs | ARC (C) | ARC (E) | BoolQ | GSM8K (S) | GSM8K (F) | HellaS | OBQA | PIQA | WinoG |
|---|---|---|---|---|---|---|---|---|---|---|---|---|
| LoRA | 8 | 16 | 2 | 56.14 | 83.88 | 83.18 | 54.36 | 54.28 | 79.44 | 45.20 | 82.10 | **75.30** |
| QLoRA | 8 | 8 | 2 | 57.08 | 83.46 | 82.48 | 53.75 | 53.90 | 79.63 | **46.00** | 82.10 | 74.59 |
| QLoRA | 8 | 4 | 2 | 54.35 | 82.41 | 82.08 | 44.50 | 44.35 | 78.82 | 44.20 | 81.50 | 73.64 |
| AdaLoRA | 8 | 16 | 2 | 52.90 | 81.99 | 81.87 | 50.57 | 50.57 | 78.65 | 45.00 | 81.34 | 73.95 |
| AdaLoRA | 8 | 16 | 5 | 53.50 | 82.25 | 82.05 | 51.00 | 50.90 | 78.75 | 45.20 | 81.40 | 74.10 |
| AdaLoRA | 8 | 8 | 2 | 52.90 | 81.86 | 82.05 | 49.96 | 49.96 | 78.65 | 44.80 | 81.34 | 74.43 |
| AdaLoRA | 8 | 8 | 5 | 53.10 | 82.00 | 82.10 | 50.20 | 50.10 | 78.70 | 45.20 | 81.38 | 74.50 |
| AdaLoRA | 8 | 4 | 2 | 51.28 | 80.98 | 80.61 | 37.83 | 38.36 | 77.36 | 42.80 | 80.74 | 72.53 |
| AdaLoRA | 8 | 4 | 5 | 51.50 | 81.10 | 80.75 | 38.00 | 38.50 | 77.40 | 43.20 | 80.78 | 72.60 |
| QR-Adaptor | 8 | 5.375 | 2 | **56.83** | **84.12** | **83.38** | **56.29** | **56.11** | **80.93** | 45.80 | **82.92** | 75.10 |

#### Observation

- **AdaLoRA's Performance with Increased Epochs**: As observed, AdaLoRA shows only slight performance improvements when training is extended from 2 to 5 epochs. Even with the increase in epochs, AdaLoRA's performance does not surpass that of LoRA, QLoRA, or QR-Adaptor at 2 epochs.

- **QR-Adaptor's Consistency**: QR-Adaptor consistently achieves superior performance across all tasks, further validating the effectiveness of our method over other adaptive rank-based approaches.

- **16-bit AdaLoRA Performance**: Notably, AdaLoRA with 16-bit precision (not quantized) still underperforms compared to LoRA and QLoRA, suggesting that the adaptive rank mechanism alone is not enough, and the integration of mixed-precision quantization is crucial.

## G.6 IMPACT OF 2-BIT QUANTIZATION AND LOFTQ ITERATIONS

We have conducted additional experiments to explore the performance of LoftQ with 2-bit quantization and its variations across different numbers of iterations.

In these experiments, we used the NF2 variant from LoftQ, based on QLoRA's NF4, to implement 2-bit quantization, since QLoRA does not natively support this low-bit quantization (as stated in the original paper and the GitHub repository). The 2-bit results in the LoftQ paper were also based on this NF2 variant. We fine-tuned the models using a 52k dataset, with the rank for LoftQ set to 16. The superscripts on LoftQ's bit-width values represent the number of LoftQ iterations, with 0 iterations considered approximately equivalent to QLoRA (since QLoRA does not provide a 2-bit quantization type).

The results of our experiments are summarized in Table 12.

**Table 12:** Performance comparison for 2-bit quantization and LoftQ iterations on LLaMA3.1-8B with 52k fine-tuning dataset. Superscripts on LoftQ bits indicate the number of initialization iterations. Accuracy is reported as %

| Method | Bit-width | MMLU | GSM8K | ARC(C) | ARC(E) | BoolQ | HellaS | OBQA | PIQA | WinoG |
|---|---|---|---|---|---|---|---|---|---|---|
| LoftQ | $2^0$ | 23.76 | 0.00 | 26.24 | 25.25 | 37.83 | 26.86 | 29.40 | 52.55 | 49.18 |
| LoftQ | $2^1$ | 24.71 | 0.00 | 25.17 | 25.25 | 37.83 | 25.73 | 29.20 | 51.58 | 49.33 |
| LoftQ | $2^5$ | 24.65 | 0.00 | 25.17 | 24.83 | 37.83 | 26.30 | 28.20 | 51.41 | 49.41 |
| LoftQ | $2^{10}$ | 24.80 | 0.00 | **26.02** | 25.25 | 37.83 | 26.53 | **29.80** | **52.83** | 48.86 |
| QR-Adaptor | 3.625 | **62.58** | 0.53 | **55.93** | **82.43** | **82.13** | **79.23** | **45.60** | **81.83** | **74.79** |

### KEY OBSERVATIONS

- **MMLU Performance**: For the MMLU dataset, which involves multiple-choice questions, models with 2-bit quantization perform at approximately 25% accuracy, which is close to random guessing. Thus, LoftQ's 2-bit quantization yields little practical improvement for MMLU on LLaMA3.1. This suggests that the performance of LoftQ with 2-bit quantization is not robust on complex tasks.

- **GSM8K Performance**: On the GSM8K dataset, LoftQ's 2-bit quantization fails to provide any meaningful performance, resulting in 0% accuracy. This highlights the challenges of quantizing LLaMA3.1 to such low precision, especially on complex question-answering tasks.

- **Common Sense Reasoning Tasks**: For simpler reasoning tasks like WinoGrande, the LoftQ 2-bit quantized models show some capacity to answer, but there is no significant difference across LoftQ's iterations, and the models still perform similarly to random guessing on most datasets.

- **QR-Adaptor Optimization**: For QR-Adaptor, we optimized based on theoretical memory savings from 4-bit quantization. Since 2-bit quantization does not reduce memory usage effectively, we used the theoretical savings in our optimization process. This optimization allowed QR-Adaptor to achieve better performance even when compared to LoftQ with 2-bit quantization.

### CONCLUSION

From these results, we observe that LoftQ's 2-bit quantization shows poor performance across the board. Even with multiple iterations (up to 10), LoftQ struggles to achieve reasonable accuracy on tasks like MMLU and GSM8K. In contrast, QR-Adaptor, with its unified optimization of both rank and bit-width during fine-tuning, consistently outperforms LoftQ and other methods.

Notably, while LoftQ's 2-bit quantization performs poorly, QR-Adaptor manages to retain much better performance by leveraging the advantages of mixed-precision quantization, making it a more effective solution for LLaMA3.1. These findings suggest that for models requiring high precision, such as LLaMA3.1, extreme quantization to 2-bit precision may not be viable, and more moderate bit-widths, as used by QR-Adaptor, provide better results.

We hope these results contribute to the ongoing discussions in the community regarding effective quantization strategies and provide further insights into the practical use of quantized models.

## H  VERSION OF LLMs

We provide the Hugging Face link of LLMs used in the experiment: LLaMA2-7B: `https://huggingface.co/NousResearch/Llama-2-7b-hf`; LLaMA2-13B: `https://huggingface.co/NousResearch/Llama-2-13b-hf`; LLaMA3.1-8B: `https://huggingface.co/meta-llama/Llama-3.1-8B`.

## I  MORE IMPLEMENTATION DETAILS

In optimizing the pruned Llama2-7B model, a carefully designed hyperparameter configuration has been implemented to strike a balance between model performance and computational efficiency. The model is fine-tuned using a learning rate of $3 \times 10^{-4}$, with a batch size of 128, divided into micro-batches of 4 to effectively manage memory limitations. Input sequences are capped at 256 tokens, and a dropout rate of 0.05 is applied to the LoRA layers, specifically targeting the query, key, value, and output projections, as well as the gate, down, and up projections. Layer-specific quantization is applied at both 4-bit and 8-bit levels, optimizing memory usage while maintaining computational accuracy. The training is performed using the paged AdamW optimizer with 32-bit precision, ensuring both stability and efficiency. These settings have been rigorously tested and refined through the Optuna framework to achieve an optimal balance between model performance and resource efficiency.

## J  MORE ABLATION

We conducted comprehensive ablation studies to evaluate the impact of initialization metrics and the sensitivity of the proposed Pareto Ranking Genetic Algorithm (PRGA) to key hyperparameters, including iteration counts and population size. These experiments aim to further substantiate the effectiveness of our proposed approach.

### J.1  GRADIENT NORMS VS. RELATIVE ENTROPY

To assess the efficacy of initialization metrics, we compared the use of gradient norms and relative entropy in quantifying layer importance for fine-tuning quantized LLMs. The experimental results are summarized in Table 13.

**Table 13:** Comparison of gradient norms and relative entropy as initialization metrics on Llama2-13B. Bold values indicate the best performance for each task. Accuracy is reported as %

| Initialization Metric | BoolQ | PIQA | HellaS | WinoG | ARC(E) | ARC(C) | OBQA | Average |
|---|---|---|---|---|---|---|---|---|
| Gradient Norms | 80.79 | 80.13 | 79.16 | 71.69 | 78.72 | 50.97 | 45.40 | 69.51 |
| Relative Entropy | **81.08** | **80.83** | **79.80** | **71.98** | **79.13** | **51.65** | **45.60** | **70.07** |

**Insights:**

- **Limitations of Gradient Norms**: Gradient norms exhibit limited variability and are prone to biases induced by quantization, which undermines their reliability as an initialization metric for quantized models.

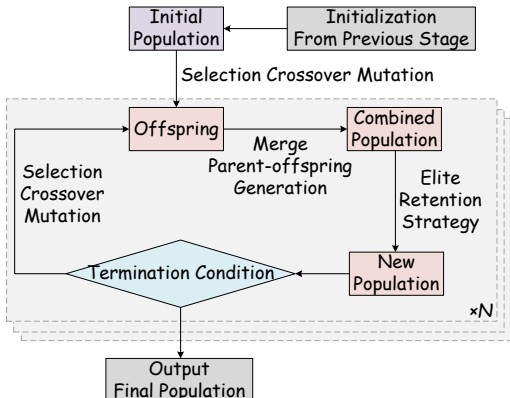

**Figure 3:** Detailed PRGA flow chart. The input is a set of solutions from the initialization, and the output is a set of Pareto front solutions containing multiple solutions.

- **Advantages of Relative Entropy**: Relative entropy captures task-specific layer importance more effectively, resulting in robust initialization and improved performance in downstream optimization.

## J.2 SENSITIVITY TO ITERATION COUNTS AND POPULATION SIZE

To analyze the sensitivity of PRGA to hyperparameters, we systematically varied the number of iterations and population sizes. Table 14 presents the results of these experiments.

**Table 14:** Sensitivity analysis of PRGA under different iteration counts and population sizes on Llama3.1-8B. Bold values indicate the best configuration.

| Iterations | Population Size | Average Improvement (%) | Total Time (min) |
|---|---|---|---|
| 5 | 3 | +0.8 | 72 |
| 5 | 5 | +1.2 | 90 |
| 10 | 5 | +1.5 | 135 |
| 5 | 20 | +1.6 | 225 |
| 10 | 20 | **+2.3** | 270 |

**Insights:**

- **Trade-offs in Population Size**: Smaller population sizes (e.g., 3) reduce computational cost but may fail to adequately explore the search space. Larger population sizes (e.g., 20) improve exploration and convergence but increase computational overhead.

- **Impact of Iteration Count**: Increasing the number of iterations improves optimization quality, as reflected in better Pareto fronts. However, the marginal benefits diminish beyond 10 iterations, indicating limited practical gains for further increases.

- **Balanced Configuration**: A population size of 5 and 5 iterations strikes a balance between performance improvement and computational efficiency. This configuration can be adjusted based on specific resource availability or performance requirements.

## K VISUALIZATION OF OPTIMIZATION STAGES

Here we visualize Stage 2: Global Exploration with PRGA (in Figure 3) and Stage 3: Local Refinement with Bayesian Optimization (in Figure 4)

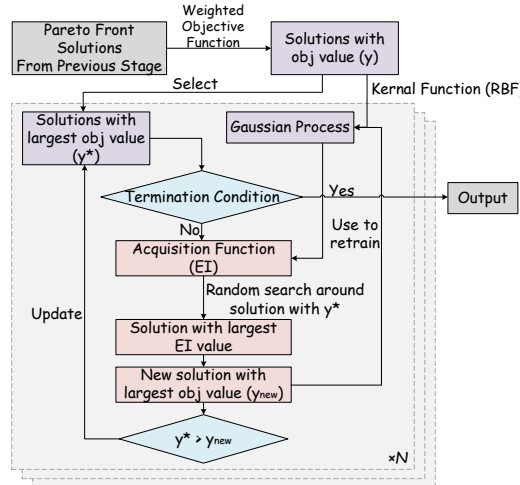

**Figure 4:** Detailed Bayesian optimization flow chart. Input is the Pareto front solution set from the global search, and output is a set of optimal solutions obtained according to the requirements.

## L    LIMITATION

Compared to previous methods, the only additional cost is time, which is mainly introduced by testing on the calibration dataset. Although the configurations optimized on different datasets have a certain degree of portability, this limitation is reduced to some extent. In addition, we are studying some approximate methods to speed up the process.

