# OpenReview forum: "Efficient Fine-Tuning of Quantized Models via Adaptive Rank and Bitwidth"
_ICLR.cc/2026/Conference — ICLR 2026 Conference Withdrawn Submission_

### Official Review · Reviewer_Y89k · 2025-10-22

**Soundness:** 2
**Presentation:** 3
**Contribution:** 2
**Rating:** 2
**Confidence:** 3

**Summary:**

This paper address the LLM quantization problem by jointly optimizing over the layer-wise bit-widths and LoRA ranks, extending the exisitng LoRA quantization algorithms to find better allocation of the limited memory resources.

**Strengths:**

The problem formulation is novel and potentially enables further improvement on LLM quantization.

**Weaknesses:**

### Methodology
- It is not clear at which stage the LoRA matrices $A, B$ are trained. Should the LoRA matrices $(A_\ell, B_\ell)$ be adapted to a fixed layer-wise memory configuration $(q_\ell ,r_\ell)$, or should a layer-wise memory configuration $q_\ell$ adapt to the fine-tuned LoRA matrices $(A_\ell, B_\ell, r_\ell)$?

- From the experiment results it seems that the proposed method does not assign heterogeneous layer-wise LoRA ranks as claimed in the main contribution. For example, Table 2 considered $r=8, 16$, Table 3 considered $r=8$, Table 5 considered $r=32$, all of which used the same LoRA rank for all layers.



### Writing
- Missing punctuation mark in line 250 and 305.

**Questions:**

- Can the authors explicitly demonstrate the improvement of each stage in Algorithm 1? For example, showing the downstream task accuracy difference after each stage.

- Since the compared baselines such as QuaRot and SpinQuant can achieve good performance on weight-activation quantization, can the author demonstrate the performance of your proposed method on weight-activation quantized models?

- How does the proposed method compare to other state-of-the-art weight-only quantization methods, such as PV-tuning with AQLM [a]?

- What kernel function is used for the Gaussian Process in stage 3? How do the authors decide the kernel function, for which it has to be suitable for the discrete value tuple $(q_\ell, r_\ell) \in \{2,4,8\} \times \mathbb{Z}$? From my perspective, it is not immediately clear how does the joint effect of bit-widths and LoRA ranks affect the similarity between different $(q_\ell, r_\ell)$. For example, consider $(q_\ell, r_\ell) = (2, 32), (4, 16), (8, 8)$, how does the adopted kernel function determine the similarity / distance of the above tuples, for the purpose of exploring the downstream performance of the quantized model?

[a] Malinovskii, Vladimir, et al. "Pv-tuning: Beyond straight-through estimation for extreme llm compression." Advances in Neural Information Processing Systems 37 (2024): 5074-5121.

---

### Official Review · Reviewer_Yvtd · 2025-10-28

**Soundness:** 3
**Presentation:** 2
**Contribution:** 2
**Rating:** 2
**Confidence:** 5

**Summary:**

This paper proposes QR-Adaptor, a framework for jointly optimizing quantization bit-width and LoRA rank per layer when fine-tuning large language models under memory constraints.
It employs a three-stage gradient-free search; (i) task-informed initialization, (ii) a Pareto-ranking genetic algorithm, and (iii) Bayesian optimization to find the optimal configuration during a single optimization process.
Experiments show that QR-Adaptor consistently achieves better accuracy–memory tradeoffs.

**Strengths:**

QR-Adaptor is the first framework to holistically allocate bit-width and LoRA rank per layer under a fixed memory budget.

Empirically, the method outperforms strong baselines and even matches 16-bit LoRA accuracy while maintaining a memory footprint comparable to 4-bit models.

The paper provides solid theoretical insights and clearly justifies the need for per-layer customization by presenting evidence of layer-wise heterogeneity. It also offers a convincing discussion on why naïve approaches can fail in discrete weight spaces.

The writing is clear and easy to follow overall.

**Weaknesses:**

1.	**Fairness of Experimental Comparison:** The experimental baselines are somewhat narrow, focusing mainly on quantization-aware LoRA-based fine-tuning methods that lack quantization-aware initialization. However, related works such as RA-LoRA [1] share a similar objective and should be included for a fair comparison. In addition, combining AdaLoRA with an initialization scheme like LoFTQ could serve as a stronger baseline to assess the true effectiveness of QR-Adaptor. It would also help to include results with more conventional 8-bit or 4-bit configurations, as the current results seem to rely on relatively high effective bit-widths, which weakens the low-bit claim of the approach.
2.	**Limited Novelty:** Layer-wise bit-width search and LoRA rank adaptation are both active research areas. This paper effectively integrates these two components, but it is unclear whether the work introduces new insights or principles that significantly advance either the LoRA-specific or quantization-specific domains. The contribution seems more like a well-engineered combination rather than a conceptual breakthrough.
3.	**Concerns on Dataset Choice:** The evaluation dataset is relatively small and limited in scope. Most of the QA benchmarks used primarily measure the model’s ability to retain accuracy compared to its full-precision counterpart, rather than demonstrating genuine fine-tuning effectiveness. To better assess fine-tuning efficacy, more challenging reasoning tasks such as GSM8K (or similar) should have been included to validate whether the proposed method improves downstream adaptation beyond accuracy preservation.
4.	**Ablation and Clarity:** The paper would benefit from deeper ablations and broader experimental coverage. For instance, the accuracy trend over a wider range of bit-widths is not clearly visible in Figure 1, making the Pareto frontier appear sharp and underexplored. Similarly, individual analyses of each component—bit-width search and rank search—would strengthen the argument that both are essential to the overall improvement.
5.	**Scalability and Search Complexity:** Since the method relies on a heuristic calibration process, a more detailed discussion of its computational complexity and scalability is needed. The current version only provides a timing analysis of the full fine-tuning stage, but not of the calibration or search phase itself. Understanding how search time scales with model size or parameter space would help clarify the practicality of deploying the method at scale.

**Questions:**

Mainly listed in the weakness. Below are the additional questions.

1. Could the authors clarify why the rank is limited to a relatively low range (8–16) instead of the standard r=64 commonly used in prior quantization-aware PEFT studies, where accuracy is known to saturate and is the optimal point of fine-tuning?

2.	Minor typo A small typo at line 250: the last sentence is missing a period (“.”).

---

### Official Review · Reviewer_w8Cb · 2025-10-31

**Soundness:** 2
**Presentation:** 2
**Contribution:** 2
**Rating:** 2
**Confidence:** 4

**Summary:**

The paper proposes QR-Adaptor, a three-stage framework that jointly allocates per-layer bit-widths (for weight quantization) and per-layer LoRA ranks under a memory budget. Stage 1 builds a task-informed initialization; Stage 2 performs Pareto-ranking genetic search to explore the discrete (rank, bit) space; Stage 3 uses Bayesian Optimization to refine along the Pareto front according to a user preference over accuracy vs. memory. The method evaluates candidate configurations by actual downstream performance on a small calibration set, explicitly avoiding proxy losses for discrete decisions. Empirically, the paper reports improvements over LoRA/QLoRA/LoftQ/AdaLoRA on LLaMA2/LLaMA3.x/Qwen models, and claims to match or exceed 16-bit LoRA while keeping a memory footprint close to 4-bit models.

**Strengths:**

* Well-motivated objective: directly optimizes the discrete accuracy–memory trade-off; the paper provides a clear argument why continuous proxies can be misleading in the presence of re-quantization discontinuities.
* Practical decomposition: a three-stage pipeline (task-informed init → PRGA → BO) is a sensible way to cover a large, discrete space under budget constraints.
* Broad model coverage: results across LLaMA2/3.x and Qwen suggest some generality (though see weaknesses on statistical rigor).
* Concrete memory formula for the double-quantized representation; useful for reproducibility and budget setting.
* Compute context: the paper at least reports per-evaluation timing (~8–9 min on LLaMA-3.1-8B) and training time comparisons for several PEFT baselines.

**Weaknesses:**

1. Initialization inconsistency: Stage-1 is defined via mutual information in the main text but via Fisher-based saliency in the appendix. Authors must resolve which criterion is actually used in each experiment and provide ablations comparing MI vs. Fisher. This is not a cosmetic issue; it changes the seed population and thereby the resulting Pareto front.
2. Compute accounting & fairness: The method’s search cost is non-negligible (population-based PRGA with multiple generations + BO, each requiring a fine-tuning/eval pass). Yet the headline wall-clock in Table 4 focuses on training without clearly adding search. For fair comparisons, report (search + training) for QR-Adaptor and equivalent hyperparameter tuning overheads for baselines. Include energy or GPU-hours.
3. Memory accounting clarity: The “exceeds 16-bit LoRA at ~4-bit memory” claim hinges on what is being counted. Please provide end-to-end memory (parameters + LoRA A/B + optimizer states/gradients during calibration/training) for all methods, not only the quantized base weights. Also clarify whether the reported memory refers to training, evaluation, or inference.
4. Statistical rigor: No confidence intervals/standard errors are reported for benchmark averages; given small deltas (often ~0.5–1.5%), significance is unclear. Provide multiple seeds, report mean±std, and specify aggregation (macro vs. micro averages, dataset sizes).
5. Generalization beyond the calibration set: The search is performed on 1,024 C4 samples (1 epoch). Show sensitivity to calibration size/source (e.g., C4 vs. task-matched data), and evaluate transferability more thoroughly (cross-dataset, cross-model) with hard numbers and confidence intervals.
6. Positioning vs. prior art: The claim of being the “first” to jointly consider per-layer rank and bit is plausibly incremental relative to the combination of mixed-precision quantization and rank-adaptation lines. A crisper Related-Work section is needed to delineate what is actually new (e.g., the search formulation and **three-stage solver**) and what is a combination of known ingredients.
7. Ablation depth: The paper includes some search sensitivity, but key ablations are missing: (i) fixed-rank / variable-bit vs. fixed-bit / variable-rank; (ii) removing each stage (init/PRGA/BO); (iii) per-layer allocation patterns and their stability across runs.

**Questions:**

* Initialization criterion: Which initialization (mutual information vs. Fisher-magnitude) was used for results in Tables 2–3? Please unify the description, release code, and add an ablation comparing both.
* Compute budget: Report total GPU-hours for the full QR-Adaptor pipeline (search + final training) and the same for each baseline including their hyperparameter tuning. How many candidate evaluations were run per model/dataset?
* Memory reporting: Provide training-time and inference-time memory separately, and explicitly include LoRA states and optimizer buffers. Do the 4-bit comparisons include these?
* Robustness to calibration data: How does performance vary if the calibration set is (a) smaller/larger than 1,024 examples, (b) sampled from the target task family instead of C4?
* Statistical significance: Please add multiple seeds and CI bars; some reported gains are within 1–2 points.
* Throughput and latency: At equal memory budgets, what are tokens/s and latency during training and inference compared to QLoRA/LoftQ/AdaLoRA?
* Release plan: The paper claims comprehensive details; please provide a code release with scripts to reproduce the search and final runs.

---

### Official Review · Reviewer_ZBwQ · 2025-11-08

**Soundness:** 1
**Presentation:** 2
**Contribution:** 1
**Rating:** 2
**Confidence:** 4

**Summary:**

This paper proposes QR-Adaptor, a framework for efficient fine-tuning of quantized large language models (LLMs). The authors argue that existing methods, which typically apply uniform bit-widths and LoRA ranks or optimize these two dimensions in isolation, are suboptimal. The paper formulates this challenge as a discrete, multi-objective optimization problem: to jointly and on a per-layer basis determine the optimal quantization bit-width and LoRA rank. The proposed solution is a three-stage, gradient-free search pipeline that leverages task-informed initialization, a Pareto-ranking genetic algorithm (PRGA), and Bayesian optimization.

**Strengths:**

- The paper addresses a highly significant and practical problem. As model compression and parameter-efficient fine-tuning (PEFT) are both critical for deploying LLMs, understanding their trade-offs and co-optimizing them is a valuable research direction.

- The paper is generally well written. The motivation is well-articulated, and the proposed three-stage optimization pipeline is explained logically, with each stage's purpose (initialization, global exploration, local refinement) well-defined.

**Weaknesses:**

- Severe Scalability Concerns: The most significant weakness is the method's tractability. The search space for the configuration seems to grow exponentially with the number of layers $L$. The experiments are confined to relatively small models (3B to 13B). The paper provides no analysis or evidence to suggest this approach is viable for current state-of-the-art models (e.g., 70B+) where $L$ is much larger, rendering the method's practical applicability on larger models questionable.

- Unconvincing Search Strategy and Cost: To manage the intractable search space, the authors rely on what appears to be an unrealistically small search configuration (e.g., Appendix J.2, Table 14, indicates a population size of 5 and 5 iterations). It is highly doubtful that such a limited search can robustly find a near-optimal solution in an exponentially large, high-dimensional space. Despite this minimal search, the method remains significantly slower than the QLORA baseline (see Table 4), suggesting a poor trade-off: the search is both computationally expensive and likely unrobust.

- Limited Methodological Novelty and Justification: The methodological contribution is limited. The 3-stage pipeline is an application of "well-established optimization techniques" (as the paper notes), combining initialization heuristics, NSGA-II (a standard genetic algorithm), and standard Bayesian Optimization. The paper does not provide a strong justification for why this specific, complex pipeline is superior to other discrete optimization strategies (e.g., simulated annealing, random search with a larger budget) for this problem.

- Lack of Deeper Analysis: The paper's core motivation is the "unaddressed interplay" and "synergistic integration" of bit-width and rank. However, the work treats this as a pure black-box optimization problem. The observation that layers have different sensitivities is not new (e.g., AdaLoRA for rank, mixed-precision quantization for bits). The paper offers no insight or analysis into the nature of this combined trade-off (e.g., why does a specific layer favor higher bits over higher rank, or vice versa?). This leaves the paper's core scientific premise—understanding the combined impact—largely unexplored.

**Questions:**

- Could the authors provide an analysis of the method's scalability? How do the search time and the final solution quality co-vary with the number of layers $L$?

- Search Robustness: Given the very small population size and iteration count used in the experiments, how robust is the final configuration $C^{*}$? If the 3-stage search is run multiple times with different random seeds, what is the variance in the performance and memory of the resulting "optimal" configurations?

- Justification of Optimization Choice: Can the authors provide a more rigorous justification for choosing a genetic algorithm (PRGA / NSGA-II) for Stage 2? What properties of this specific optimization landscape (e.g., high epistasis, many local optima) make it more suitable than other, potentially simpler, combinatorial optimization methods?

---

### Author Response · Authors · 2025-11-13
**Factual Discrepancies in Reviews**

We thank the reviewers for their time.

However, upon reading the reviews for our paper, it became immediately apparent that the four "reject" ratings are not based on good-faith academic disagreement, but on a critical failure to read the submitted paper.

The reviews are rife with demonstrably false claims that are directly contradicted by the text. The core justifications for rejection rely on asserting that key components are "missing" when they are explicitly detailed in the manuscript. Some specific examples are (and many are even fake claims).

Claim: Harder tasks like GSM8K are missing.

Fact: GSM8K results are in many tables, like Table 2 (Section 4.2) and Appendix G.

Claim: The method does not use per-layer ranks.

Fact: This is the entire point of our method. The reviewer clearly mistook our method for the baselines. (Section 2, Table 1).

Claim: The GP kernel is not specified.

Fact: It is specified in Appendix E (Table 6).

Claim: There is no ablation of the method's three stages.

Fact: Section 4.4 ("Ablation Study") and Appendix J are dedicated to this.

Reviewers have a fundamental responsibility to read and evaluate the work they are assigned. The nature of these errors is so fundamental, so systemic in overlooking explicit content, that it goes far beyond what "limited time" or "oversight" can explain. This work has gone through several rounds of revision over the last year. In earlier submissions, the paper usually received borderline or weak-accept scores.

Numerous signs strongly suggest that some reviewers are relying entirely on AI tools to automatically generate peer reviews, rather than fulfilling their fundamental responsibility of personally reading and evaluating manuscripts.

We strongly protest this.

This is a gross disrespect to the authors. It is a flagrant desecration of the reviewer's sacred duty. It fundamentally undermines the integrity of the entire peer-review process.

Given that the reviews are not based on the actual content of our paper, we have decided to withdraw the submission.

We leave this comment so that future readers of the OpenReview page are aware that the items described as "missing" are already present in the submitted manuscript. These negative reviews for this submission are factually unsound and do not reflect the content of the paper. We cannot and will not accept an assessment that is not based on the work we actually submitted.

---

### Note · Authors · 2025-11-13

I have read and agree with the venue's withdrawal policy on behalf of myself and my co-authors.